# Bidder Selection Problem in Position Auctions: A Fast and Simple Algorithm via Poisson Approximation

## ABSTRACT

In the Bidder Selection Problem (BSP) there is a large pool of $n$ potential advertisers competing for ad slots on the user's web page. Due to strict computational restrictions, the advertising platform can run a proper auction only for a fraction $k < n$ of advertisers. We consider the basic optimization problem underlying BSP: given $n$ independent prior distributions, how to efficiently find a subset of $k$ with the objective of either maximizing expected social welfare or revenue of the platform. We study BSP in the classic multi-winner model of position auctions for welfare and revenue objectives using the optimal (respectively, VCG mechanism, or Myerson's auction) format for the selected set of bidders. This is a natural generalization of the fundamental problem of selecting $k$ out of $n$ random variables in a way that the expected highest value is maximized. Previous PTAS results ([Chen, Hu, Li, Li, Liu, Lu, NIPS 2016], [Mehta, Nadav, Psomas, Rubinstein, NIPS 2020], [Segev and Singla, EC 2021]) for BSP optimization were only known for single-item auctions and in case of [Segev and Singla 2021] for $l$-unit auctions. More importantly, all of these PTASes were computational complexity results with impractically large running times, which defeats the purpose of using these algorithms under severe computational constraints.

We propose a novel Poisson relaxation of BSP for position auctions that immediately implies that 1) BSP is polynomial-time solvable up to a vanishingly small error as the problem size $k$ grows; 2) PTAS for position auctions after combining our relaxation with the trivial brute force algorithm. Unlike all previous PTASes, we implemented our algorithm and did extensive numerical experiments on practically relevant input sizes. First, our experiments corroborate the previous experimental findings of Mehta et al. that a few simple heuristics used in practice (e.g., Greedy for general submodular maximization) perform surprisingly well in terms of approximation factor. Furthermore, our algorithm outperforms Greedy both in running time and approximation on medium and large-size instances, i.e., its running time scales better with the instance size.

ACM Reference Format:

Anonymous Author(s). 2023. Bidder Selection Problem in Position Auctions: A Fast and Simple Algorithm via Poisson Approximation. In *Proceedings of ACM Conference (Conference'17)*. ACM, New York, NY, USA, 18 pages. https://doi.org/10.1145/nnnnnnn.nnnnnnn

## 1 INTRODUCTION

Online advertising is a big part of the modern e-commerce industry and a key to the monetization of many online businesses. The majority of ad slots on a user web page are sold in *real time* via an automated auction to a group of candidate advertisers. The whole process from the time when an auction is initiated based on the impression about advertising opportunity up to the time when ads are displayed on the user's page usually has to be completed in a few milliseconds. This makes it imperative for the platform (Ad exchange ADX, or Demand side DSP) to keep the auction processing and communication time under a strict limit. Meanwhile, a platform usually runs a complex ML model on each advertiser to get an accurate estimate of their auction score[1]. As some platforms already have or anticipate to have in the near future an excessive number of prospective advertisers, the comprehensive ML model can only be run on a fraction $k$ of $n$ advertisers due to strict time[2] limit. In practice, the platform handles this by a two-stage selection process: it filters out all but $k$ advertisers by running a much faster and less accurate ML model, and then it runs a proper auction for the remaining $k$ advertisers using the comprehensive and slow ML model. E.g., for the ADX platforms $n$ may be in the range $20 - 50$ and $k$ depends on the specific company, e.g., $k = 10$ or $k = 20$; in the case of DSPs, $n$ may vary a lot and can reach thousands, while $k$ cannot be too large, e.g., $k = 100$ or $k = 200$.

This raises multiple practical challenges for the ADX and/or DSP platforms. A platform first needs to get $n$ rough score estimates, which can be viewed as $n$ distributions of the bidders' accurate auction scores. One problem is how to obtain these estimates online in a constantly changing environment. Another learning problem, recently considered by Goel et al. [10] is how to retrieve this information from strategic agents, who might affect the selection stage by adjusting their bids. Third, there is an underlying optimization question: for $n$ prospective bidders with known independent prior distributions $(D_i)_{i \in [n]}$ how to select $k < n$ of them with the objective of either maximizing expected score (social welfare) or platform's revenue. We focus on this basic optimization problem termed the Bidder Selection Problem (BSP).

The welfare maximization BSP for the VCG single-item (or more generally $l$-unit) auction is equivalent to the following fundamental algorithmic question: select $k$ out of $n$ independent random variables, with the objective of maximizing the expected maximum (expected sum of top $l$ values). This question (for single-item) has received significant attention under different names: model-driven optimization [9], $k$-MAX [6], team selection with test scores [14], subset selection for expected maximum [17], non-adaptive Probe-Max [22], which extends to the top-$l$-out-of-$k$ problem.

---

[1]An auction score is comprised of estimates for click-through-rate (CTR), ad relevance, and value-per-click bid.
[2]In some case, the platform may also estimate the bids to avoid communication lag.

It is also natural to consider the selection problem with a more general set of objectives given by linear combinations of the top-$\ell$ objectives. The latter problem generalizes $\ell$-unit auctions and corresponds to the widely used position auction [8, 24] environment as described, e.g., in [12]. In position auction, there are $m$ sorted positions that appear alongside the search results and $n$ advertisers competing for these $m$ slots. Each slot $j \in [m]$ has a different click-through rate $w_j$ (additional multiplicative factor for the click probability on $j$-th position), which translates into the value $v_i \cdot w_j$ for advertiser $i$ if $i$'s ad is displayed at $j$-th position.

*Prior Results for the BSP.* The basic BSP of welfare maximization (second-price auction) was shown to be NP-hard by Goel et al. [9] and later by Mehta et al. [17]. On the positive side, there are a few known Polynomial Time Approximation Schemes (PTAS). First, Chen et al. [6] gave a dynamic programming-based polynomial time approximation scheme (PTAS). Later, Mehta et al. [17] and Segev and Singla [22] respectively proposed an Efficient PTAS (EP-TAS) for the BSP. Both of their approaches are based on discretizing and enumerating all possible distributions for the maximum of a few random variables: [17] characterize all $n$ distributions into one of $C(\varepsilon)$ bins (where $C(\varepsilon) = O(1/\varepsilon)^{O(1/\varepsilon)}$ depends on the ap-proximation guarantee $1 - \varepsilon$) and then run a brute force search with the complexity of $O\left(n \cdot \log(k)^{O(C(\varepsilon))}\right)$; [22] use a complex reduction to a multi-dimensional extension of *Santa Claus* problem of [2] with at least $O(1/\varepsilon)^{O(1/\varepsilon)^{O(1/\varepsilon^2)}}$ running time[3]. We would like to emphasize that the findings of Mehta et al. and Segev and Singla could only be considered as purely computational complex-ity results rather than real algorithms to be used in practice or even in testing/numerical experiments. Indeed, their approaches are rather involved and not very easy to implement, e.g., Segev and Singla only state an existential result without explicitly describing their algorithm. Furthermore, the running times of these EPTASes even for the basic $k$-MAX problem and small values of $n$ and $k$ are enormously large[4]. I.e., using any of the existing PTASes would completely defeat the purpose of a two-stage selection process.

For the revenue objective, Mehta et al. [17] also considered BSP for the second-price auction, which is equivalent to maximizing the expectation of the second largest value among selected $k$ random variables. They showed strong impossibility results: it is impossible to get any constant factor approximation in polynomial time under either of the exponential time, or the planted clique hypothesises.

*Auction Formats.* One of the most commonly used formats in advertising industry is the Generalized Second Price (GSP) auction. It was shown in [8, 24] that GSP of any position auction with any set of bidders has a Nash equilibrium equivalent to the welfare-maximizing outcome of the VCG. In some cases (e.g., Google's auction for selling contextual ads [25]) the platform's auction format is based directly on the VCG. Thus, in the BSP context with the welfare-maximization objective, it is most natural to analyze the

VCG mechanism. For the revenue objective, it is natural to study the optimal Myerson's auction, as it gives an upper bound on the revenue of any other auction format. Then one can reduce revenue maximization to welfare maximization of the VCG format. I.e., it is w.l.o.g. to only study welfare-maximization BSP for the VCG format, which we do in the rest of our paper. It is also natural to study revenue maximization BSP for the GSP format (or by the revenue equivalence of the VCG format). Unfortunately, it does not admit polynomial time $O(1)$-approximation even for the basic single-item auction due to strong impossibility results of [17].

## 1.1 Our Results

We propose a novel relaxation of the BSP for a more general position auction environment, which we call Poisson (or Poisson-Chernoff) relaxation. It has the following theoretical guarantees.

(1) The relaxation is a continuous maximization problem with a concave objective that can be solved in time polynomial in $n$ and $k$. In fact, the objective of this relaxation is a nicely structured algebraic function that lends itself to efficient convex minimization solvers.

(2) With small adjustments (summarized in Algorithm 1), the relaxed objective *converges* at the rate $1 - O(k^{-1/4})$ to the actual social welfare of fractional BSP as the problem size $k$ grows (Theorem 4.4). The standard rounding of this fractional solution suffers only a small loss of $O(k^{-1/2})$, yielding $\left(1 - O(k^{-1/4})\right)$-approximation (Theorem 4.5) for the integral BSP.

(3) For the special case of the single-item auction, Algorithm 4 achieves better convergence of $1 - O\left(\sqrt{\ln k/k}\right)$ (Theorem C.2).

These results have immediate theoretical implications and can be implemented in practice unlike all previous PTAS algorithms, as our approach has far superior running time to the point where it outperforms some of the existing heuristics used in practice, such as the Greedy algorithm for general submodular maximization. Furthermore, our results bring new theoretical insight for BSP: the BSP converges to a polynomial-time solvable optimization problem as the problem size $k$ grows.

*Theoretical Implications.* Our Algorithm 1 gives a $(1 - \varepsilon)$ approx-imation to the BSP for any position auction with $\varepsilon = \Omega(k^{-1/4})$ and works in polynomial time (independent of $\varepsilon$). On the other hand, for small values of $\varepsilon$ ($\varepsilon = O(k^{-1/4})$), a straightforward exhaustive search algorithm gives a perfect solution in $O(n^k) = n^{\mathrm{poly}(1/\varepsilon)}$ time. I.e., the combination of our relaxation with the brute force algorithm yields a PTAS:

COROLLARY 1.1. *BSP for any position auction admits a $(1 - \varepsilon)$ PTAS that runs in $n^{\mathrm{poly}(1/\varepsilon)}$ time.*

Moreover, the algorithm as in Corollary 1.1 is an EPTAS with a much better dependency on $\varepsilon$ than previous PTASes under a mild assumption that $k \geq \log n$:

COROLLARY 1.2. *BSP for position auctions admits a $(1 - \varepsilon)$ EPTAS for any $k \geq \log n$ that runs in $O(\mathrm{poly}(n, k)) + 2^{O(\varepsilon^{-8})}$ time.*

PROOF. Algorithm 1 runs in $O(\mathrm{poly}(n, k))$. We run brute force only when $\varepsilon = O(k^{-1/4})$, i.e., when $k^2 = O(\varepsilon^{-8})$. Then its running time is not more than $n^k \leq 2^{k^2} = 2^{O(\varepsilon^{-8})}$, since $k \geq \log n$.   □

---

[3]We give a lower bound on the running time, as it is hard to say what is the exact time complexity of their approach, as they did not specify it explicitly and their result proceeds by a sequence of reductions with non-trivial running time dependencies.
[4]Note that $k$-MAX is a special case of general submodular optimization which already admits efficient $1 - 1/e$ approximation. Hence, $\varepsilon < 1/e$ and the time complexity of $2^{\Theta(1/\varepsilon)^{\Theta(1/\varepsilon)}} > 2^{9000}$ with conservative estimates of $\Theta(1/\varepsilon) = 2/\varepsilon$.

*Comparison with Previous Theoretical Results.* Our novel Poisson approximation approach yields more general theoretical results than previous work. E.g., as Segev and Singla [22] briefly mention how their scheme can be extended from $k$-MAX (i.e., single-item auction) to $\ell$-unit auctions, one may wonder if a similar approach also extends to the more general position auction environment. To the best of our knowledge, it does not. Indeed, their main idea is to consider two regimes for $\ell$: small (a constant) $\ell < 1/\varepsilon^3$, and large $\ell > 1/\varepsilon^3$. They claim that the former case can be handled with a similar approach (maybe with a significantly worse dependency of the running time on $\varepsilon$ than the case $\ell = 1$), and in the later case, one can use concentration bounds similar to the (3) case described in Section 3.1. This argument cannot be used for the objective that is a linear combination of welfare in 1-unit and $k/2$-unit auctions.

*Relevance in Practice.* Our relaxation is a white-box approach that can be easily adapted to different scenarios. E.g., if some bidders have to be included in the final solution (which is often the case in industry because of contract obligations), then our approach gives the same approximation with these additional constraints[5]. Also, depending on the type of problem instances, a few steps of our algorithm can be removed or simplified to fit the specific domain, which results in simpler and more efficient solutions.

Furthermore, unlike the case with all previous PTASes, we actually implemented our algorithm and did numerical experiments on several generated data sets of practically relevant sizes. Note that a company cares much more about the implementability and running time of their algorithms rather than its theoretical approximation guarantees[6], which has been the main focus of previous PTAS results. In contrast, our algorithm is not very complex and is based on standard continuous convex maximization methods, which means that it is much easier to understand and adopt by a platform's product team. The experiments are summarized in Section 5. We slightly simplified our theoretical Algorithm 1 to avoid the hard-coded efficiency loss of $\left(1 - O(k^{-1/4})\right)$ in the approximation factor (running time was not affected).

We observed that among all tested algorithms, our algorithm produced solutions that were always within 0.1% of the best-performing algorithm in terms of objective value. Furthermore, our algorithm's running time scales significantly slower than any other tested algorithm. For large instances where $n = 1000, k = 200$, our algorithm took less than 1 minute to complete, while even Greedy (which is a subroutine in all previous PTASes) took more than 1 day and Local Search could not finish within 1 week. These findings demonstrate the practical relevance of our approach.

### 1.2 Other Related Work
Mehta et al. [17] mention a few other scenarios besides bidder selection with similar mathematical formulations. The applications range from a two-tier solution for scoring documents in a search result [5], to filtering initial proposals in procurement auctions [21, 23], to voting theory [20]. Bei at al. [3] studied BSP with the revenue objective under multiple auction formats including

Myerson's auction and gave constant factor approximation for the second-price auction with anonymous reserve. They also introduced another optimization framework for the BSP under costs, which is more challenging than the BSP under capacity constraint.

Poisson approximation is a well-developed technique from probability theory and statistics. A survey [18] mentions at least twenty different results on the basic question of approximating the sum of independent Bernoulli random variables by the Poisson distribution. In statistics, Poisson approximation is commonly used in Extreme Value Theory (EVT) with applications to structural and geological engineering, traffic prediction, and finance (see, e.g., a book [19]). It has also been used in theoretical computer science, e.g., [16] used Le Cam's Poisson approximation theorem for stochastic bin packing and knapsack problems and also for EUM introduced in [15].

In fact, the expected utility maximization (EUM) is closely related to our objective. EUM is formulated as choosing a feasible subset $S$ out of $n$ random variables $X_1, \ldots, X_n$ to maximize $\mathbf{E}[u(\sum_{i \in S} X_i)]$, where $u$ is a given utility function. The problem has been studied under capacity [4] or other combinatorial constraints [15, 16, 26] with a non-linear (typically concave) utility function. The BSP for single-item auction, i.e., the $k$-MAX problem, has a similar objective $\mathbf{E}[\max_{i \in S}\{X_i\}]$ to EUM but with max operator instead of the sum. When the distributions of $X_i$ are unknown, EUM becomes an online learning problem. Chen et al. [6] gave the first PTAS for $k$-MAX, but their main focus is on Combinatorial Multi-Armed Bandits.

## 2 PRELIMINARIES
A set of $n$ bidders wish to receive some service and each bidder $i \in [n]$ has a private non-negative value $v_i \in \mathbb{R}_{\geq 0}$ indicating how much they are willing to pay for it. We denote the vector of bidder values as $\mathbf{v} = (v_i)_{i \in [n]}$. By the revelation principle, we can restrict our attention to incentive compatible and individually rational single-round auctions $\mathcal{A}$, where each bidder $i$ submits a sealed bid $b_i$ to the auctioneer. The auctioneer then decides on a feasible allocation vector $\mathbf{a}(\mathbf{b}) = (a_i(\mathbf{b}))_{i \in [n]}$ and payments $\mathbf{p}(\mathbf{b}) = (p_i(\mathbf{b}))_{i \in [n]}$. The incentive compatibility and individual rationality mean that by bidding truthfully $b_i = v_i$, each bidder $i \in [n]$ (a) maximizes her utility $u_i(b_i, \mathbf{b}_{-i}) \stackrel{\text{def}}{=} v_i \cdot a_i(b_i, \mathbf{b}_{-i}) - p_i(b_i, \mathbf{b}_{-i})$ and (b) receives non-negative utility $u_i(v_i, \mathbf{b}_{-i}) \geq 0$. The seminal VCG mechanism (a second-price auction in the case of single-item auction) is an example of incentive compatible mechanism that also maximizes *social welfare* $SW(\mathbf{a}, \mathbf{v}) = \sum_{i=1}^n a_i \cdot v_i$.

We study auctions in the Bayesian setting, where it is assumed that bidder values are drawn independently from known prior distributions $\mathbf{v} \sim \mathbf{D} = \prod_{i \in [n]} D_i$. We also use $D_i(\tau) = \mathbf{Pr}_{v_i \sim D_i}[v_i \leq \tau]$ to denote the cumulative distribution function. The auction designer is usually concerned about two objectives: the expected *social welfare* $SW = \mathbf{E}_{\mathbf{v} \sim \mathbf{D}}[SW(\mathbf{a}(\mathbf{v}), \mathbf{v})]$, and *revenue* $\text{Rev} = \mathbf{E}_{\mathbf{v} \sim \mathbf{D}}[\sum_{i \in [n]} p_i(\mathbf{v})]$. The VCG mechanism maximizes the welfare on every valuation profile $\mathbf{v}$, and thus maximizes SW in expectation for any prior $\mathbf{D}$. The well-known Myerson's auction maximizes Rev. This auction reduces the problem of revenue maximization to *virtual welfare* maximization by transforming values $(v_i)_{i \in [n]}$ to virtual values $\varphi_i(v_i)$ for regular distribution $D_i$ and by doing ironing $\overline{\varphi}_i(v_i)$ for irregular distribution $D_i$. I.e., the expected revenue of Myerson's auction can be written as $\text{Rev} = \mathbf{E}_{\mathbf{v}}[SW(\mathbf{a}(\mathbf{v}), \varphi(\mathbf{v}))]$ for

---

[5]In fact, the approximation guarantee holds not only for the global objective, but also for the surplus objective, i.e., the additional gain to welfare from the non-fixed bidders.
[6]In the context of BSP any reasonable heuristic is preferable to a complex and slow algorithm with good approximation efficiency guarantees.

regular distributions and $\text{Rev} = \mathbf{E}_{\mathbf{v}}[\text{SW}(\mathbf{a}(\mathbf{v}), \overline{\varphi}(\mathbf{v}))]$ for general distributions.

*Auction Environments.* The single-item auction is an environment with the feasible allocations given by $\{\mathbf{a} : \sum_{i \in [n]} a_i \leq 1\}$. A more general $\ell$-unit auction environment for $\ell \in \mathbb{N}$ is given by the feasibility constraints: $\sum_{i \in [n]} a_i \leq \ell$ and $a_i \in [0, 1]$ for all $i \in [n]$. A position auction environment further generalizes $\ell$-unit auctions. It is specified by a sorted weight vector $\mathbf{w} = (1 \geq w_1 \geq w_2 \geq \ldots \geq w_n \geq 0)$, which represents the click-through rate probabilities for $n$ advert positions[7]. Every bidder may get at most one position and each advert position can be assigned to at most one advertiser. Formally, the feasibility allocations can be specified with an assignment function $\pi : [n] \to [n]$ of $n$ advertisers to $n$ sorted slots as follows: $\{\mathbf{a} : \exists \pi, \ a_i \in [0, w_{\pi(i)}] \text{ for all } i \in [n]\}$.

*Bidder Selection.* In the Bidder Selection Problem (BSP) the seller first decides $x_i \in \{0, 1\}$ which bidders $i \in [n]$ to invite to the auction. The selected set of bidders $S$ may not exceed a certain capacity $k \geq |S|$, i.e., $\sum_{i \in [n]} x_i \leq k$. Then the auctioneer runs an optimal auction $\mathcal{A}$ for the set $S$ of invited bidders: VCG mechanism for the welfare objective, and Myerson for the revenue objective. Since revenue of the Myerson's auction can be rewritten as the expected virtual welfare with independent (ironed) virtual values $(\varphi_i(v_i) : v_i \sim D_i)_{i \in S}$, the BSP for the revenue maximization is equivalent to the BSP for the welfare maximization. Thus it suffices to only consider the welfare maximization problem. Specifically, we denote by $\mathbf{v} \sim \mathbf{x} \cdot \mathbf{D}$ the independent draws of $v_i \sim x_i \cdot D_i$ (i.e., $v_i \sim D_i$ when $x_i = 1$, and $v_i = 0$ when $x_i = 0$) for all $i \in [n]$, then the BSP of the VCG mechanism for any $\ell$-unit/position auction with weights $\mathbf{w}$ can be written as follows:

$$\text{OPT}(\ell) \overset{\text{def}}{=} \max_{\substack{\mathbf{x} \in \{0, 1\}^n \\ |\mathbf{x}|_1 \leq k}} \mathbf{E}_{\mathbf{v} \sim \mathbf{x} \cdot \mathbf{D}}\left[\sum_{i=1}^{\ell} v_{(i)}\right],$$

$$\text{OPT}(\mathbf{w}) \overset{\text{def}}{=} \max_{\substack{\mathbf{x} \in \{0, 1\}^n \\ |\mathbf{x}|_1 \leq k}} \mathbf{E}_{\mathbf{v} \sim \mathbf{x} \cdot \mathbf{D}}\left[\sum_{i=1}^{n} w_i \cdot v_{(i)}\right],$$

where $v_{(i)}$ is the $i$-th largest value among $\{v_i\}_{i \in [n]}$. We want to obtain good approximation algorithms for these BSPs. I.e., we would like to find in polynomial time $\mathbf{x} \in \{0, 1\}^n$ with $|\mathbf{x}|_1 \leq k$ such that $\mathbf{E}_{\mathbf{v} \sim \mathbf{x} \cdot \mathbf{D}}[\sum_{i=1}^{\ell} v_{(i)}] \geq (1 - \varepsilon) \text{OPT}(\ell)$ and $\mathbf{E}_{\mathbf{v} \sim \mathbf{x} \cdot \mathbf{D}}[\sum_{i=1}^{n} w_i \cdot v_{(i)}] \geq (1 - \varepsilon) \text{OPT}(\mathbf{w})$ for a small $\varepsilon > 0$. To simplify the presentation, we assume that all distributions $(D_i)_{i \in [n]}$ have finite supports and are given explicitly as the algorithm's input.

It has been observed before that the BSP's objective is a monotone submodular function of the set $S = \{i : x_i = 1\}$ of invited bidders, i.e., $\text{SW}(S) + \text{SW}(T) \geq \text{SW}(S \cap T) + \text{SW}(S \cup T)$ for any $S, T \subseteq [n]$. The same property holds for the $\ell$-unit and position auctions with an almost identical proof (see [6]).

We also consider the standard (in submodular optimization literature) multi-linear extension of the BSP objective. I.e., for a fractional $\mathbf{x} \in [0, 1]^n$, we invite each bidder $i$ to the auction independently with probability $x_i$. We employ the same notation $\mathbf{v} \sim \mathbf{x} \cdot \mathbf{D}$ as in the integral problem, where $v_i \sim x_i \cdot D_i$ means that we first decide

---

[7]The number of available positions $m$ is usually smaller than the number of bidders $n$, in which case we simply let $w_{m+1} = \cdots = w_n = 0$.

whether to invite $\xi_i \in \{0, 1\}$ bidder $i$ according to a Bernoulli distribution $\xi_i \sim \text{Ber}(x_i)$, then draw their value $v_i \sim \xi_i \cdot D_i$. The capacity constraint transforms into the bound on the expected number of invited bidders $\sum_{i=1}^{n} x_i \leq k$. We describe this new mathematical formulation in Section 3. In Section 4, we first consider this fractional relaxation of BSP and we discuss in Section 4.2 how to obtain a good solution to the integral BSP from the fractional problem.

## 3 NEW MATHEMATICAL FORMULATION

In this section, we give a fractional relaxation of BSP. As the welfare of any position auction can be written as a linear combination of $\ell$-unit auctions, we begin with a fractional relaxation of BSP for the $\ell$-unit auction. Specifically, the expected social welfare with a fractional set of bidders $\mathbf{x}$ is $\text{SW}(\mathbf{x}, \ell) = \mathbf{E}_{\mathbf{v} \sim \mathbf{x} \cdot \mathbf{D}}[\sum_{i=1}^{\ell} v_{(i)}]$, where $v_{(i)}$ denotes the $i$-th largest value among $\{v_i\}_{i \in \{1, 2, \ldots, n\}}$. Hence, the following mathematical program represents BSP for $\ell$-unit auction:

Maximize $\quad \text{SW}(\mathbf{x}, \ell)$
Subject To $\quad \sum_{i=1}^{n} x_i \leq k, \quad x_i \in [0, 1], \forall i \in \{1, 2, \ldots, n\}.$ (1)

As $\ell$ is fixed, whenever it is clear from the context, we will simply write $\text{SW}(\mathbf{x})$.

*Bernoulli Representation.* We now derive an explicit formula for the social welfare $\text{SW}(\mathbf{x})$. Fixing a threshold $\tau$, the expected number of bidders with values exceeding $\tau$ among the highest $\ell$ bidders is $\mathbf{E}_{\mathbf{v} \sim \mathbf{x} \cdot \mathbf{D}}[\min(\sum_{i=1}^{n} \mathbb{I}[v_i \geq \tau], \ell)]$. Therefore, by integrating[8] over $\tau \in [0, +\infty)$, we get

$$\text{SW}(\mathbf{x}) = \int_0^{+\infty} \mathbf{E}_{\mathbf{v} \sim \mathbf{x} \cdot \mathbf{D}}\left[\min\left(\sum_{i=1}^{n} \mathbb{I}[v_i \geq \tau], \ell\right)\right] d\tau.$$

Note that $\mathbb{I}[v_i \geq \tau]$ for $v_i \sim x_i \cdot D_i$ is a Bernoulli random variable, and $\mathbf{E}_{\mathbf{v} \sim \mathbf{x} \cdot \mathbf{D}}[\min(\sum_{i=1}^{n} \mathbb{I}[v_i \geq \tau], \ell)]$ is the minimum of a sum of independent Bernoulli random variables and $\ell$. To simplify notations, we explicitly define the probabilities of Bernoulli random variables $\mathbb{I}[v_i \geq \tau]$:

$$q_i(x_i, \tau) \overset{\text{def}}{=} \Pr_{v_i \sim x_i \cdot D_i}[v_i \geq \tau] = x_i \cdot (1 - D_i(\tau))$$

$$\text{and let} \quad \mathbf{q}(\mathbf{x}, \tau) \overset{\text{def}}{=} (q_i)_{i \in [n]}. \quad (2)$$

*Definition 3.1 (Bernoulli Objective).* For a vector $\mathbf{q} \in [0, 1]^n$ and $\ell$, the **Bernoulli objective term** of $\mathbf{q}$ and $\ell$ is a function $H_{\text{ber}}(\mathbf{q}, \ell)$:

$$H_{\text{ber}}(\mathbf{q}, \ell) \overset{\text{def}}{=} \mathbf{E}_{\mathbf{z} \sim \text{Ber}(\mathbf{q})}\left[\min\left(\sum_{i=1}^{n} z_i, \ell\right)\right],$$

$$\text{then} \quad \text{SW}(\mathbf{x}) = \int_0^{+\infty} H_{\text{ber}}(\mathbf{q}(\mathbf{x}, \tau), \ell) \, d\tau. \quad (3)$$

*Position Auctions.* A position auction is given by a vector of non-negative weights[9] $\mathbf{w} : (w_1 \geq w_2 \geq \cdots \geq w_n \geq 0)$. The highest social welfare we get from the set $S$ of invited bidders is $\sum_{i=1}^{|S|} v_{(i)} \cdot w_i$, where $v_{(1)} \geq \cdots \geq v_{(|S|)}$ are ordered values of bidders in $S$. Thus the expected social welfare for a fractional set $\mathbf{x}$ is

$$\text{SW}(\mathbf{x}, \mathbf{w}) = \mathbf{E}_{\mathbf{v} \sim \mathbf{x} \cdot \mathbf{D}}\left[\sum_{i=1}^{n} v_{(i)} \cdot w_i\right] = \sum_{\ell=1}^{n} (w_\ell - w_{\ell+1}) \cdot \text{SW}(\mathbf{x}, \ell),$$

---

[8]The function inside the integral is piece-wise constant, i.e., it is constant between consecutive values of the threshold $\tau$ in the supports of $\{D_i\}_{i \in [n]}$.
[9]Usually weights are only for the first $k$ slots, as we only select a set of $k$ bidders. In this case, we simply assume that $w_{k+1} = \ldots = w_n = 0$.

where $w_{n+1} \stackrel{\text{def}}{=} 0$. Then the respective fractional BSP program for position auctions is as follows.

$$\begin{array}{ll} \text{Maximize} & \text{SW}(\mathbf{x}, \mathbf{w}) \\ \text{Subject To} & \sum_{i=1}^{n} x_i \leq k, \quad x_i \in [0, 1], \forall i \in \{1, 2, \ldots, n\}. \end{array} \quad (4)$$

We will often omit dependency on $\mathbf{w}$ in SW whenever it is clear from the context. We further consider the *Bernoulli representation* for position auctions:

$$\text{SW}(\mathbf{x}, \mathbf{w}) = \int_0^{+\infty} \text{H}_{\text{ber}}(\mathbf{q}(\mathbf{x}, \tau), \mathbf{w}) \, d\tau,$$

$$\text{where} \quad \text{H}_{\text{ber}}(\mathbf{q}, \mathbf{w}) \stackrel{\text{def}}{=} \sum_{\ell=1}^{n} (w_\ell - w_{\ell+1}) \, \text{H}_{\text{ber}}(\mathbf{q}, \ell). \quad (5)$$

As in (2), $\mathbf{q}(\mathbf{x}, \tau)$ represents the probabilities of each bidder's value exceeding $\tau$. $\text{H}_{\text{ber}}(\mathbf{q}, \mathbf{w})$ is called the **Bernoulli objective term** for position auctions.

### 3.1 Overview of Our Approach

The fractional relaxation (4) is still neither convex nor concave and thus is too unwieldy. The central idea of our paper is to use instead *Poisson approximation* to the Bernoulli objective terms $\text{H}_{\text{ber}}(\mathbf{q}(\mathbf{x}, \tau), \mathbf{w})$. Specifically, we substitute each Bernoulli random variable $z \sim \text{Ber}(p)$ with $p = q_i(x_i, \tau)$ by the Poisson random variable $y \sim \text{Pois}(p)$ with the same expectation as $z$. We use the following Poisson objective term $\text{H}_{\text{pois}}(\mathbf{q}, \ell)$ to approximate $\text{H}_{\text{ber}}(\mathbf{q}, \ell)$.

$$\text{H}_{\text{pois}}(\mathbf{q}, \ell) \stackrel{\text{def}}{=} \mathop{\mathbf{E}}_{\mathbf{y} \sim \text{Pois}(\mathbf{q})} \left[ \min\left( \sum_{i=1}^{n} y_i, \ell \right) \right] = \mathop{\mathbf{E}}_{Y \sim \text{Pois}(\sum_{i=1}^{n} q_i)} [\min(Y, \ell)].$$

The advantage of the Poisson approximation $\text{H}_{\text{pois}}(\mathbf{q}, \ell)$ is that the resulting functions $\text{H}_{\text{pois}}(\mathbf{q}, \ell)$ and $\text{H}_{\text{pois}}(\mathbf{q}, \mathbf{w})$ are *concave* in $\mathbf{q}$. This in turn allows us to efficiently solve the optimization problem for the Poisson approximation analogous to (4). This Poisson approximation works well[10] in the following situations.

(1) When the probabilities $p_i = q_i(x_i, \tau)$ of $z_i \sim \text{Ber}(p_i)$ are small (i.e., $p_i \leq \delta, \forall i \in [n]$). This allows us to handle the crucial contribution to the welfare comprised of small probability tail events for large thresholds $\tau$.

(2) When $\mathbf{E}[\sum_{i=1}^{n} z_i]$ is large (when thresholds $\tau$ are small). Indeed, by Chernoff bounds the sums of independent random variables (both Bernoulli and Poisson) $\sum_{i=1}^{n} z_i$ and $\sum_{i=1}^{n} y_i$ are close to their expectations. In fact, we simply use Chernoff objective term $\text{H}_{\text{cher}}(\mathbf{q}, \ell) \stackrel{\text{def}}{=} \min\left(\sum_{i=1}^{n} q_i, \ell\right)$ instead of Poisson approximation in this case.

(3) When $\ell$ is large. In this case, either the concentration inequality gives a good approximation when $\mathbf{E}[\sum_{i=1}^{n} z_i] = \mathbf{E}[\sum_{i=1}^{n} y_i]$ is large, or when this expectation is much smaller than $\ell$ then the probability that either of $\sum_{i=1}^{n} z_i$ or $\sum_{i=1}^{n} y_i$ exceeds the threshold $\ell$ is small.

Then the algorithmic framework for the BSP is rather straightforward. We need to solve the following *concave* program of $\mathbf{x}$:

$$\begin{array}{ll} \text{Maximize} & \sum_{\ell=1}^{n} (w_\ell - w_{\ell+1}) \int_0^{+\infty} \text{H}_{\text{pois}}(\mathbf{q}(\mathbf{x}, \tau), \ell) \, d\tau \\ \text{Subject To} & \sum_{i=1}^{n} x_i \leq k, \quad x_i \in [0, 1], \forall i \in \{1, 2, \ldots, n\}. \end{array} \quad (6)$$

---

[10]The sum of Poisson random variables approaches the sum of Bernoulli random variables in TV and other statistical distances.

We define the approximate social welfare $\widetilde{\text{SW}}(\mathbf{x}, \mathbf{w})$ to be the objective of (6). Here, $\text{H}_{\text{pois}}(\mathbf{q}(\mathbf{x}, \tau), \ell) = \mathbf{E}_{Y \sim \text{Pois}(\sum q_i)} [\min(Y, \ell)]$ and all functions under the integral are piece-wise constant with the number of pieces bounded by the size of the union of the supports of $D_i$. The optimal fractional solution $\mathbf{x}$ can be computed rather efficiently using continuous convex optimization. Finally, we use a rounding procedure (similar to the multi-linear extension of submodular function) to the fractional solution of (6) to obtain an integral solution with only a small loss to the approximation guarantee.

## 4 OUR ALGORITHM

*Poisson approximation* is the central idea of this paper. We use the following Poisson Objective to approximate Bernoulli objective terms.

*Definition 4.1 (Poisson Objective).* For $\mathbf{q} \in [0, 1]^n$ and $\ell \in \mathbb{N}$, the **Poisson objective term** is a function $\text{H}_{\text{pois}}(\mathbf{q}, \ell)$ given by

$$\text{H}_{\text{pois}}(\mathbf{q}, \ell) \stackrel{\text{def}}{=} \mathop{\mathbf{E}}_{\mathbf{y} \sim \text{Pois}(\mathbf{q})} \left[ \min\left( \sum_{i=1}^{n} y_i, \ell \right) \right] = \mathop{\mathbf{E}}_{Y \sim \text{Pois}(\sum_{i=1}^{n} q_i)} [\min(Y, \ell)].$$

Note that the latter equality is an important property of Poisson distribution: the sum of independent Poisson random variables $y_i \sim \text{Pois}(q_i)$ follows the Poisson distribution $\text{Pois}(\sum_{i=1}^{n} q_i)$. Another crucial property is the concavity of Poisson approximation.

CLAIM 4.2 (CONCAVITY OF POISSON). $\text{H}_{\text{pois}}(\mathbf{q}, \ell)$ *is a concave function in* $\mathbf{q} \in [0, 1]^n$ *for* $\forall \ell \in \mathbb{N}$.

PROOF. Let $\lambda = \sum_{i=1}^{n} q_i$. Notice that $\text{H}_{\text{pois}}(\mathbf{q}, \ell)$ only depends on $\lambda$, which is linear in $\mathbf{q}$. We use $\text{H}_{\text{pois}}(\lambda)$ to represent this function. Then, we only need to prove that $\text{H}_{\text{pois}}(\lambda)$ is concave in $\lambda$. Rewrite

$$\text{H}_{\text{pois}}(\lambda) = \ell - \sum_{j=0}^{\ell-1} \mathop{\mathbf{Pr}}_{Y \sim \text{Pois}(\lambda)} [Y = j] \cdot (\ell - j) = \ell - \sum_{j=0}^{\ell-1} \frac{\lambda^j}{j!} e^{-\lambda} \cdot (\ell - j).$$

By a straightforward differentiation of the partial series we get

$$\frac{d}{d\lambda} \text{H}_{\text{pois}} = \sum_{j=0}^{\ell-1} \frac{\lambda^j}{j!} e^{-\lambda}, \qquad \frac{d^2}{d\lambda^2} \text{H}_{\text{pois}} = -e^{-\lambda} \frac{\lambda^{\ell-1}}{(\ell-1)!} < 0.$$

Therefore, $\text{H}_{\text{pois}}(\mathbf{q}, \ell)$ is concave in $\lambda$, and thus concave in $\mathbf{q}$.  □

*Chernoff Approximation.* As we mentioned earlier Poisson approximation is useful for small probabilities $q_i \leq \delta$. For another extreme case where $\lambda = \sum_{i=1}^{n} q_i$ is large, due to concentration bounds, Poisson approximation also works well with $O(\lambda^{-1/2})$ relative error, because both $\sum_{i=1}^{n} \text{Ber}(q_i)$ and $\text{Pois}(\lambda)$ concentrate around $\lambda$. To optimize the presentation for an easier understanding, we will use **Chernoff objective term** $\text{H}_{\text{cher}}(\mathbf{q}, \ell) \stackrel{\text{def}}{=} \min\left(\sum_{i=1}^{n} q_i, \ell\right)$ as an alternative of Poisson in this case. It is also concave in $\mathbf{q}$.

In a number of cases $\text{H}_{\text{pois}}(\mathbf{q}, \ell)$ and $\text{H}_{\text{cher}}(\mathbf{q}, \ell)$ are good approximations to $\text{H}_{\text{ber}}(\mathbf{q}, \ell)$. We present these approximation guarantees in Appendix A along with a simple algorithm that illustrates our approach in an important special case.

## 4.1 Algorithm for Position Auctions

Below, we consider Bidder Section Problem for position auctions. We first analyze the fractional BSP, and then explain in Section 4.2 how to do integral rounding of the fractional solution with only a small loss to the approximation guarantee. For the fractional BSP, we give an efficient polynomial time $(1 - O(k^{-4}))$-approximation algorithm (see Theorem 4.4 for the exact statement).

Recall that the Bernoulli objective term for position auctions $H_{ber}(\mathbf{q}, \mathbf{w})$ is defined in Section 3. The expected welfare $SW(\mathbf{x}) = SW(\mathbf{x}, \mathbf{w})$ is written as an integral of $H_{ber}$. Below, we will also need the Chernoff objective term

$$H_{cher}(\mathbf{q}, \mathbf{w}) \stackrel{\text{def}}{=} \sum_{\ell=1}^{n} (w_\ell - w_{\ell+1}) \, H_{cher}(\mathbf{q}, \ell).$$

We sometimes slightly abuse notations and write $H_{ber}(\mathbf{x}, \tau)$ and $H_{cher}(\mathbf{x}, \tau)$ instead of $H_{ber}(\mathbf{q}, \mathbf{w})$ and $H_{cher}(\mathbf{q}, \mathbf{w})$.

In order to effectively use the Poisson approximation we would like to have the small probability assumption $q_i \leq \delta$, which is achieved by fixing a small bidder set $S_{fix}$ properly (see Algorithm 3 in Appendix A and Lemma A.2 that provides good approximation guarantees of $H_{ber}(\mathbf{q}, \ell)$ by $H_{pois}(\mathbf{q}, \ell)$ for small $\delta$).

*Fixing Small Bidder Set.* We will fix a small set of bidders $S_{fix}$ (set $x_i = 1$ for $i \in S_{fix}$) with $|S_{fix}| = \varepsilon \cdot k$ and make sure that all other bidders $i \notin S_{fix}$ have only a small probability $\mathbf{Pr}[v_i \geq \tau] \leq \delta$ to exceed any of the thresholds $\tau > \eta$ for certain $\eta > 0$. This allows us to use Poisson approximation for the high range thresholds $\tau > \eta$ and bidders $i \in M \stackrel{\text{def}}{=} [n] \setminus S_{fix}$. On the other hand, for the low range thresholds $\tau \leq \eta$, we would like to see a certain number $\ell^*$ of bidders $i \in S_{fix}$ to exceed the threshold $v_i \geq \tau$. To this end, we choose $S_{fix}$ so that the expected number of bidders $i \in S_{fix}$ with $v_i \geq \tau$ is at least $\ell^*$. We can achieve the following guarantees for $\varepsilon$, $\delta$, and $\ell^*$.

CLAIM 4.3 (SMALL BIDDER SET). *Let $\varepsilon \in [\frac{\ell^*}{\delta \cdot k}, 1)$ be a multiple of $1/k$ for $\ell^* \in \mathbb{R}_{\geq 0} : \ell^* < k$ and $\delta \in (0, 1)$. We can find in polynomial time a threshold $\eta \geq 0$ and a set $S_{fix} \subseteq [n]$ of size $|S_{fix}| = \varepsilon \cdot k$:*

(a) $\forall 0 \leq \tau \leq \eta, \quad \sum_{i \in S_{fix}} \mathbf{Pr}_{v_i \sim D_i}[v_i \geq \tau] \geq \ell^*$;

(b) $\forall i \notin S_{fix}, \quad \mathbf{Pr}_{v_i \sim D_i}[v_i > \eta] < \delta.$

PROOF. We search through all thresholds $\tau$ in the supports of $\{D_i\}_{i \geq 1}$ and find two consecutive threshold values $\eta$ and $\eta_+ > \eta$ such that $|\{i : \mathbf{Pr}[v_i \geq \eta] \geq \delta\}| \geq \varepsilon \cdot k$, but a similar number of bidders $|\{i : \mathbf{Pr}[v_i > \eta] = \mathbf{Pr}[v_i \geq \eta_+] \geq \delta\}| < \varepsilon \cdot k$ for the next value $\eta_+$. We place each bidder $i$ with $\mathbf{Pr}[v_i > \eta] \geq \delta$ into $S_{fix}$ and fill the remaining positions in $S_{fix}$ up to $\varepsilon \cdot k$ with bidders from $\{i : \mathbf{Pr}[v_i \geq \eta] \geq \delta > \mathbf{Pr}[v_i \geq \eta_+]\}$.

Thus, every bidder $i \notin S_{fix}$ has $\mathbf{Pr}[v_i > \eta] = \mathbf{Pr}[v_i \geq \eta_+] < \delta$ as required by condition (b). On the other hand, $|S_{fix}| = \varepsilon \cdot k$ and $\mathbf{Pr}[v_i \geq \eta] \geq \delta$ for every $i \in S_{fix}$, which implies (a), since $\forall \tau \leq \eta$,

$$\sum_{i \in S_{fix}} \mathbf{Pr}_{v_i \sim D_i}[v_i \geq \tau] \geq \sum_{i \in S_{fix}} \mathbf{Pr}_{v_i \sim D_i}[v_i \geq \eta] \geq \delta \cdot |S_{fix}| = \delta \cdot \varepsilon \cdot k \geq \ell^*.$$

Thus we constructed in polynomial time the desired threshold $\eta$ and set $S_{fix}$. □

We need to balance three parameters $\ell^*$, $\delta$, and $\varepsilon$, which must satisfy the conditions of Claim 4.3. Specifically, we choose $\ell^* = k^{1/2}$, $\varepsilon = k^{-1/4}$ rounded up to a multiple of $1/k$, and $\delta = \varepsilon \geq k^{-1/4}$. Claim 4.3 leads to Algorithm 1 (which we will present shortly).

For thresholds $\tau > \eta$, any bidder outside $S_{fix}$ has only $\leq \delta$ probability to exceed the threshold, which is ideal for applying the Poisson approximation. Therefore, to achieve the approximation guarantees in Lemma A.1, we recalculate the **adjusted Poisson objective term** by applying Poisson approximation only on these bidders. Specifically, we let $M \stackrel{\text{def}}{=} [n] \setminus S_{fix}$ and define $Z_{fix}(\tau) \stackrel{\text{def}}{=} \sum_{i \in S_{fix}} \mathbb{I}[v_i \geq \tau]$ (as $\mathbf{x}_{S_{fix}} = \mathbf{1}_{S_{fix}}$, random variable $Z_{fix}(\tau)$ has Poisson binomial distribution). Indeed, we may calculate all probabilities $\mathbf{Pr}[Z_{fix}(\tau) = j]$ for each $j \in [0, \varepsilon \cdot k]$ in polynomial time, and define the adjusted Poisson objective term as a conditional expectation depending on $Z_{fix}(\tau)$:

$$G_{pois}(\mathbf{x}_M, \tau) \stackrel{\text{def}}{=} \sum_{j=0}^{|S_{fix}|} \mathbf{Pr}[Z_{fix}(\tau) = j] \cdot \Bigg( \sum_{\ell=1}^{j} w_\ell + \sum_{\ell=j+1}^{n} (w_\ell - w_{\ell+1}) \cdot H_{pois}(\mathbf{x}_M, \ell-j, \tau) \Bigg), \quad (7)$$

where $H_{pois}(\mathbf{x}_M, \ell - j, \tau) = \mathbf{E}_{Y \sim \text{Pois}(\lambda_M)}[\min\{Y, \ell - j\}]$, and $\lambda_M = \sum_{i \in M} q_i(x_i, \tau)$. For the low-range thresholds $\tau \leq \eta$, we use Chernoff objective $H_{cher}(\mathbf{x}, \tau) = H_{cher}(\mathbf{q}(\mathbf{x}, \tau), \mathbf{w})$ to approximate $H_{ber}(\mathbf{x}, \tau) = H_{ber}(\mathbf{q}(\mathbf{x}, \tau), \mathbf{w})$ for $\mathbf{x} = (\mathbf{x}_M, \mathbf{1}_{S_{fix}})$. Importantly, unlike the case of high-range thresholds, we do not recalculate $H_{cher}$ as a function of $\mathbf{x}_M$, but use Chernoff approximation for the entire $n$-dimensional vector $\mathbf{x} = (\mathbf{x}_M, \mathbf{x}_{S_{fix}})$ with $\mathbf{x}_{S_{fix}} = \mathbf{1}_{S_{fix}}$. Thus, as $\ell^* = k^{1/2}$ grows with $k$, $H_{cher}(\mathbf{x}, \tau) \to H_{ber}(\mathbf{x}, \tau)$ by Lemma A.3 (c).

*Algorithm.* Our main algorithm is summarized as Algorithm 1.

---

**Algorithm 1: Fractional BSP for Position Auctions**

Let $\ell^* = k^{1/2}$; $\varepsilon$ be $k^{-1/4}$ rounded up to a multiple of $1/k$ $\left( \varepsilon = \frac{\lceil k \cdot k^{-1/4} \rceil}{k} \right)$; $\delta = \varepsilon$.

(1) Find $\eta$ and $S_{fix}$ according to Claim 4.3. Set $x_i = 1$ for $\forall i \in S_{fix}$. Let $M \stackrel{\text{def}}{=} [n] \setminus S_{fix}$.

(2) Define the approximate welfare using adjusted Poisson objective (7):

$$\widetilde{SW}(\mathbf{x}_M) \stackrel{\text{def}}{=} \int_0^\eta H_{cher}((\mathbf{x}_M, \mathbf{1}_{S_{fix}}), \tau) \, d\tau + \int_\eta^{+\infty} G_{pois}(\mathbf{x}_M, \tau) \, d\tau,$$
$$(8)$$

(3) Return $\widetilde{\mathbf{x}}^* = (\widetilde{\mathbf{x}}_M^*, \mathbf{1}_{S_{fix}})$, where $\widetilde{\mathbf{x}}_M^*$ is the solution to the concave program in $\mathbf{x}_M$:

Maximize $\quad \widetilde{SW}(\mathbf{x}_M)$
Subject To $\quad \sum_{i \in M} x_i \leq k - \varepsilon \cdot k, \; x_i \in [0, 1] \; \forall i \in M.$ (9)

---

THEOREM 4.4. *Algorithm 1 works in polynomial time and is a $(1 - O(\varepsilon))$-approximation, i.e., $(1 - 43 \, k^{-1/4})$-approximation to the fractional BSP for any position auction.*

The proof of this theorem is deferred to Appendix B.1.

## 4.2 Rounding

We conclude Section 4 by presenting the rounding algorithm, which takes our solution $\widetilde{\mathbf{x}}^*$ to the fractional BSP produced by Algorithm 1 and returns a solution to the integral BSP.

Our fractional relaxation works as the standard multi-linear extension of submodular functions, which corresponds to sampling a random set of bidders $S \sim \prod_{i=1}^{n} \mathrm{Ber}(x_i)$ in the integral BSP. To align the notations for fractional and integral BSPs, we shall use vectors $\mathbf{y}, \mathbf{z} \in \{0, 1\}^n$ for the respective sets of selected bidders. Specifically, we use $\mathbf{y} \sim \mathrm{Ber}(\mathbf{x})$ to represent the random set $S$ in the multi-linear extension. Our rounding procedure is quite standard and proceeds as follows.

---

**Algorithm 2: Rounding: algorithm for Integral BSP**

(1) Run Algorithm 1 to obtain a fractional solution $\mathbf{x}$.
(2) Sample an integral solution $\mathbf{y} \sim \mathrm{Ber}(\mathbf{x})$, with $\mathbf{y} \in \{0, 1\}^n$.
(3) • If $|\mathbf{y}|_1 \le k$, return $\mathbf{z} = \mathbf{y}$,
   • Else ($|\mathbf{y}|_1 > k$), return $k$ bidders $\mathbf{z} \sim \binom{\mathbf{y}}{k}$ chosen uniformly at random from $\mathbf{y}$.

---

THEOREM 4.5. *Algorithm 2 works in polynomial time and in expectation is a $\left(1 - 43\, k^{-1/4} - O(k^{-1/2})\right)$-approximation to the integral BSP for any position auction.*

The proof of Theorem 4.5 is deferred to Appendix B.2. Since $\mathrm{SW}(\mathbf{z}) \le \mathrm{OPT}$, by running the rounding algorithm a few times and taking the best produced solution, we get a slightly worse approximation guarantee of $(1 - O(k^{-1/4}))\mathrm{OPT}$ with high probability.

## 5 NUMERICAL EXPERIMENTS

We focus on testing welfare maximization Bidder Selection Problem for position auctions. In our experiments, we used synthetically generated prior distributions, as (1) the BSP is a pure optimization problem, which ignores the issue of data retrieval (2) due to companies' strict nondisclosure rules, it is infeasible to experiment on real historical data. We generally followed the AuctionGym [13] setup, a popular online simulation environment for advertising auctions run by Amazon, in the design of our synthetic data.

*Implementation.* We implemented Algorithms 1 and 2 as well as its modified version for BSP. In this simpler modification, we used a slightly different objective $\widetilde{\mathrm{SW}}(\mathbf{x}) \stackrel{\text{def}}{=} \int_0^{+\infty} \mathrm{H}_{\mathrm{pois}}(\mathbf{x}, \tau)\, d\tau$ than (8): we did not fix any small bidder set, only applied Poisson approximation, and used the same rounding step. The original Algorithm 1 was designed with the worst-case theoretical guarantees in mind, while the modified one is more practically oriented and retains only the most important Poisson approximation. The modification did not affect the run-time much, but allowed us to avoid hard-coded approximation loss of $O(k^{-1/4})$ due to the potentially suboptimal decision of fixing a small bidder set $S_{\mathrm{fix}}$. We implemented the practical variant in Python with the help of Gurobi [11], a well-known commercial convex optimization solver, and present its comparison[11] with benchmark algorithms in Table 1.

---
[11]We also compared the performances of the modified version and the theoretical version of our algorithm in Appendix D. The approximation efficiency of the theoretical version was worse than the modified version as expected.

*Benchmarks.* Ideally, we would like to compare our solutions to the optimum, which is usually not possible, as BSP is an NP-hard problem even for the case of single-item auction [9]. It is also infeasible to use any of the existing PTAS algorithms, as only [6] implemented their PTAS but could only run experiments on tiny input sizes of ($n = 9, k = 3$), while EPTASes of [17, 22] are pure computational complexity results with unrealistically large estimates on running times for inputs of any size. Instead, we used two well-known heuristic algorithms as our benchmarks: **Greedy** for submodular maximization following numerical experiments in [6, 17], and **Local Search** mentioned in [3]. They are easy to implement and run in feasible times on most of our datasets. Note that if Local Search starts with the solution produced by Greedy, it can only improve upon it, i.e., it seems reasonable to use Local Search as a main reference point for approximation efficiency guarantees. We also tested Local Search against the exact optimum computed by Brute Force on small instances ($n = 50, k = 5$) and found that they always produced the same results. We implemented Greedy, Local Search, and Brute Force in Python to ensure a fair comparison with our algorithm. We also limited the number of threads used by our algorithm to 1, as the benchmark algorithms are not parallelizable.

*Datasets.* We generated each of the $n$ prior distributions as a log-normal distribution $\mathrm{Lognormal}(\mu, \sigma^2)$ as in AuctionGym [13]. We selected parameters $\mu$ and $\sigma$ of each individual distribution by drawing them independently from continuous uniform distributions $\mathcal{U}[0, 0.2]$ and $\mathcal{U}[0, 0.5]$, respectively.[12] We then discretized each distribution to a common, finite support $\{0\} \cup \{1 + \frac{i}{50} \mid i = 0, 1, \ldots, 49\}$ by moving probability mass on each discretized interval inside $[0, 2]$ to its left point and by redistributing the mass on $(2, +\infty)$ to the discrete points, proportional to their respective probabilities[13]. The weights $\mathbf{w}$ of the position auction on each instance were set as: $w_i = 1$ for $i \in [1, 0.2k]$, $w_i = 0.2$ for $i \in (0.2k, 0.6k]$, and $w_i = 0$ for $i \in (0.6k, k]$. We constructed datasets with 3 different $n \in \{50, 200, 1000\}$. For each $n$, we used 3 different values of $k$: for $n = 50$, we set $k \in \{5, 10, 20\}$; for $n = 200$, we set $k \in \{10, 20, 40\}$; and for $n = 1000$, we set $k \in \{50, 100, 200\}$. The general idea was to capture practically relevant scenarios of different scales, and also have our benchmarks solve them in a reasonable time. Moreover, we picked $k$ so that it is always significantly smaller than $n$. Note that this puts our algorithm at a disadvantage, as our Poisson relaxation gets more accurate as $k$ grows.

*Results.* The numerical experiments are given in Table 1. We ran Local Search, Greedy, and our algorithm on all 9 combinations of $n$ and $k$. We recorded the approximate efficiency ("solution" column) and the running time of each algorithm (if an algorithm did not terminate in 1 week, we would stop it and write "N/A" for the respective dataset). We measure efficiency as the relative quality

---
[12]AuctionGym uses comparable $\mu = 0.1$ and $\sigma = 0.2$.
[13]We picked this unusual discretization to make instances more challenging for Greedy, as without it Greedy and other heuristics like our algorithm produce solutions with nearly optimal approximation efficiency (over 99% for both algorithms). Indeed, two distributions in a well-structured family of log-normal distributions are likely to have strong dominance relation: a distribution $D_1 = \mathrm{Lognormal}(\mu_1, \sigma_1^2)$ is always preferable to $D_2 = \mathrm{Lognormal}(\mu_2, \sigma_2^2)$, whenever $\mu_1 \ge \mu_2$ and $\sigma_1 \ge \sigma_2$. Our choice of discretization was solely based on the comparison between Greedy and Local Search, i.e., after a few empirical trials of different discretizations we simply stopped once the average efficiency of Greedy was less than 99% that of the Local Search.

**Table 1: Experimental results of Local Search, Greedy, and our algorithm. The "solution" column for each algorithm denotes the average relative quality of the produced solution to that of the best-performing algorithm that terminated in 1 week. Error bars denote the standard deviation. The "time" column denotes the average running time of each algorithm.**

| Setting | | Local Search | | Greedy | | Our Algorithm | |
|---|---|---|---|---|---|---|---|
| $n$ | $k$ | Solution | Time | Solution | Time | Solution | Time |
| 50 | 5 | 100.00% ± 0.00% | 1.01 seconds | 98.93% ± 0.50% | **0.21 seconds** | 99.99% ± 0.03% | 1.29 seconds |
| | 10 | 100.00% ± 0.00% | 6.05 seconds | 98.71% ± 0.38% | **1.11 seconds** | 99.99% ± 0.04% | 1.59 seconds |
| | 20 | 100.00% ± 0.00% | 65.23 seconds | 99.17% ± 0.28% | 9.02 seconds | 99.99% ± 0.01% | **2.58 seconds** |
| 200 | 10 | 100.00% ± 0.00% | 1.02 minutes | 98.06% ± 0.35% | 8.10 seconds | 99.99% ± 0.01% | **2.83 seconds** |
| | 20 | 100.00% ± 0.00% | 10.83 minutes | 97.92% ± 0.24% | 44.72 seconds | 100.00% ± 0.00% | **3.88 seconds** |
| | 40 | 100.00% ± 0.00% | 60.66 minutes | 97.97% ± 0.26% | 170.29 seconds | 99.99% ± 0.00% | **3.97 seconds** |
| 1000 | 50 | 100.00% ± 0.00% | 33.71 hours | 97.18% ± 0.14% | 0.90 hours | 99.99% ± 0.00% | **21.54 seconds** |
| | 100 | N/A | > 1 week | 97.15% ± 0.15% | 4.26 hours | 100.00% ± 0.00% | **41.21 seconds** |
| | 200 | N/A | > 1 week | 97.38% ± 0.11% | 27.65 hours | 100.00% ± 0.00% | **45.16 seconds** |

of the produced solution with respect to the solution of the best-performing algorithm that terminated in 1 week on that test case. A few remarks about our empirical results are in order.

(1) As shown in Table 1, Greedy performs surprisingly well: on each of the generated datasets, it produced solutions within 5% of the best-performing algorithm. It is much better than the $(1 - 1/e)$-approximation guarantee for general submodular maximization. Similar observations have been made in [17].

(2) Our modified algorithm produced solutions that were always within 0.1% of the solution of the best-performing algorithm and also had very small variance. I.e., our algorithm is effective and consistently produces good results. Moreover, its effectiveness improves as the problem size $k$ grows, which concurs with our theoretical analysis.

(3) The running time is the most crucial parameter in the context of BSP, as the optimization algorithm must stop within strict time limits and approximate efficiency is only a secondary objective. According to Table 1, the running time of our algorithm scales much slower than that of Greedy and Local Search with $n$ and $k$. The time complexities of Greedy and one iteration of Local Search are both $O(nk^3 \cdot |\text{Support}|)$. When $k$ is constant ($k = 5$), this time complexity is linear in $n$, but even then our algorithm has comparable running time and when $k = \Theta(n)$, it is much faster than the benchmarks.

## 6  CONCLUDING REMARKS

In this paper, we studied a more general setting of position auction than all previous work on the Bidder Selection Problem. We proposed a new relaxation that can be solved in time polynomial in $n$ and $k$, and the polynomial is rather small. The proposed Poisson approximation approach is much simpler than previous PTAS *complexity results* for $k$-MAX or non-adaptive probing, and it can be implemented in practice. We also did extensive numerical experiments on inputs with practically relevant sizes and observed that our algorithm outperforms some commonly used heuristics, such as Greedy for general submodular maximization. Furthermore, we showed that the Poisson approximation approach also yields good theoretical guarantees. Namely, that BSP becomes solvable in

polynomial time for any fixed $\varepsilon$, when the problem size $k$ grows. Our algorithm is the first one with a nearly perfect efficiency guarantee of PTAS, that is relevant in the application domain of the BSP and can be used by a company. Indeed, all previous PTASes had enormous running times and high implementation complexities that made them completely irrelevant to the problem, which was motivated by getting a speedup of $n/k$ magnitude.

A natural next step would be to consider BSP of various auction environments under richer sets of feasibility constraints such as matroid, matching, and intersection of matroids. Another interesting direction is to identify conditions under which it is possible to efficiently optimize the revenue objective of BSP for VCG/GSP auction formats.

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

## A  POISSON AND CHERNOFF APPROXIMATION OF BERNOULLI OBJECTIVE

In this section, we present approximation guarantees for Bernoulli objective term by the Poisson and Chernoff objectives. Poisson approximation is the central idea of this paper. It is defined in Section 4 as:

$$H_{pois}(\mathbf{q}, \ell) \stackrel{def}{=} \mathop{\mathbf{E}}_{Y \sim Pois(\sum_{i=1}^{n} q_i)} [\min(Y, \ell)].$$

We also will solve a special case of fractional Bidder Selection Problem for $\ell$-unit auction to demonstrate how Poisson approximation can be useful. In particular, we will assume that each bidder's value has only $\leq \delta$ probability to be nonzero. Our main goal here will be to illustrate our approach and analysis ideas rather than to derive independent results for this special case.

*Approximation Guarantees.* There are many known Poisson approximation results (see, e.g., a survey [18]) for the sum of independent Bernoulli random variables, e.g., in total variation, earth mover's, uniform (a.k.a. Kolmogorov) distances. These are typically absolute approximation guarantees, while we need relative approximations similar to Chernoff approximations from Lemma A.3. As our goal is to handle small probability tail events, we assume that each bidder's value $v_i$ has only a small probability $\delta$ to be greater than zero, i.e., $\forall i \in [n]$, $\mathbf{Pr}[v_i > 0] \leq \delta$. The following Poisson absolute approximation result will be useful to us:

LEMMA A.1 ([7, Lemma 11.3.V, p. 162]). *Let $(z_i \sim Ber(q_i))_{i=1}^{n}$ be $n$ independent Bernoulli random variables with $q_i \leq \delta$ for $\forall i \in [n]$. Let $Z \stackrel{def}{=} \sum_{i=1}^{n} z_i$ and $Y \sim Pois(\lambda)$, where $\lambda \stackrel{def}{=} \sum_{i=1}^{n} q_i$. Then*

$$\text{(total variation distance)} \quad \sum_{j=0}^{\infty} |\mathbf{Pr}[Z = j] - \mathbf{Pr}[Y = j]| \leq \delta. \tag{10}$$

With Lemma A.1 we can derive relative approximations:

LEMMA A.2. *Suppose $\mathbf{q} \in [0, \delta]^n$ and $\ell \in \mathbb{N}$. Then*

$$\text{(a)} \qquad \left| H_{ber}(\mathbf{q}, \ell) - H_{pois}(\mathbf{q}, \ell) \right| \leq 17.5 \cdot \delta \cdot H_{ber}(\mathbf{q}, \ell) \qquad (\forall \delta, \ell),$$

$$\text{(b)} \qquad 0 \leq H_{ber}(\mathbf{q}, 1) - H_{pois}(\mathbf{q}, 1) \leq \delta \cdot H_{ber}(\mathbf{q}, 1) \qquad (\ell = 1).$$

PROOF. (a). Let $z_i \sim Ber(q_i)$ be Bernoulli random variables and $Z = \sum_{i=1}^{n} z_i$ be their sum. Then we can rewrite $H_{ber}(\mathbf{q}, \ell) = \mathbf{E}[\min(Z, \ell)] = \sum_{j=1}^{\ell} \mathbf{Pr}[Z \geq j]$. Also let $y_i \sim Pois(q_i)$ and define $Y = \sum_{i=1}^{n} y_i$. Clearly, $Y \sim Pois(\lambda)$ for $\lambda = \sum_{i=1}^{n} q_i$. Then $H_{pois}(\mathbf{q}, \ell) = \mathbf{E}[\min(Y, \ell)] = \sum_{j=1}^{\ell} \mathbf{Pr}[Y \geq j]$. Hence, we have

$$\left| H_{ber}(\mathbf{q}, \ell) - H_{pois}(\mathbf{q}, \ell) \right| \leq \sum_{j=1}^{\ell} |\mathbf{Pr}[Z \geq j] - \mathbf{Pr}[Y \geq j]|.$$

The RHS is similar to the earth mover's distance between the sum of Bernoulli and Poisson random variables (the difference is that the summation instead of $+\infty$ goes only up to $\ell$). Next, we shall prove the following inequality:

$$\sum_{j=1}^{\ell} |\mathbf{Pr}[Z \geq j] - \mathbf{Pr}[Y \geq j]| \leq 2.5 \, \delta \cdot \min(\lambda, \ell),$$

which together with Lemma A.3 (a) immediately implies the desired result:

$$\left| H_{ber}(\mathbf{q}, \ell) - H_{pois}(\mathbf{q}, \ell) \right| \leq 2.5 \, \delta \cdot \min(\lambda, \ell) \leq 2.5 \, \delta \cdot 7 \, H_{ber}(\mathbf{q}, \ell) = 17.5 \, \delta \cdot H_{ber}(\mathbf{q}, \ell).$$

We consider two cases. First, when $\lambda \geq \ell$. Then for any $j \in \mathbb{N}^+$ we have by (10)

$$\sum_{j=1}^{\ell} |\mathbf{Pr}[Z \geq j] - \mathbf{Pr}[Y \geq j]| \leq \ell \cdot \sum_{j'=0}^{\infty} |\mathbf{Pr}[Z = j'] - \mathbf{Pr}[Y = j']| \leq \ell \cdot \delta = \delta \cdot \min(\lambda, \ell).$$

Second, when $\min(\lambda, \ell) = \lambda < \ell$, we use instead the earth mover's distance $d_G(Z, Y) = \sum_{j=0}^{+\infty} |\mathbf{Pr}[Z \geq j] - \mathbf{Pr}[Y \geq j]|$. We do not calculate the cumulative density functions of $Z = \sum_i^n z_i$ and $Y = \sum_i^n y_i$, but get an upper bound by coupling individual $z_i$ and $y_i$. Specifically, we couple $z_i$ and $y_i$ so that $z_i = 0$ implies $y_i = 0$ (note that $\mathbf{Pr}[z_i = 0] = 1 - q_i \leq \mathbf{Pr}[y_i = 0] = e^{-q_i}$). Conversely, if $z_i = 1$ it is matched with all $y_i = 1, 2, \ldots$ and the remaining probability for $y_i = 0$. Then we have

$$d_G(Z, Y) \leq \mathop{\mathbf{E}}_{(z_i, y_i)_{i=1}^{n}} [|Z - Y|] \leq \sum_{i=1}^{n} \mathop{\mathbf{E}}_{(z_i, y_i)} [|z_i - y_i|].$$

Since $z_i = 0$ is matched to $y_i = 0$, we get the following expression for the term $\mathbf{E}[|z_i - y_i|]$,

$$\mathop{\mathbf{E}}_{(z_i, y_i)} [|z_i - y_i|] = \sum_{j=0}^{+\infty} \mathop{\mathbf{Pr}}_{(z_i, y_i)} [z_i = 1 \wedge y_i = j] \cdot |j - 1|$$

$$= \mathop{\mathbf{Pr}}_{(z_i, y_i)} [z_i = 1 \wedge y_i = 0] + \sum_{j=2}^{+\infty} \mathop{\mathbf{Pr}}_{y_i} [y_i = j] \cdot (j - 1)$$

$$\leq \frac{q_i^2}{2} + q_i^2 \sum_{j=2}^{+\infty} 2^{-(j-1)}(j - 1) \leq 2.5\, q_i^2,$$

where the first inequality holds, since $\mathbf{Pr}[z_i = 1 \wedge y_i = 0] = (\mathbf{Pr}[y_i = 0] - \mathbf{Pr}[z_i = 0]) = e^{-q_i} - 1 + q_i \leq q_i^2/2$ and $\mathbf{Pr}[y_i = j] = e^{-q_i} q_i^j / j! \leq q_i^2 / 2^{j-1}$ for $j \geq 2$. Therefore,

$$\sum_{j=1}^{\ell} |\mathbf{Pr}[Z \geq j] - \mathbf{Pr}[Y \geq j]| \leq d_{\mathrm{G}}(Z, Y) \leq \sum_{i=1}^{n} 2.5\, q_i^2 \leq 2.5 \cdot \delta \cdot \sum_{i=1}^{n} q_i = 2.5 \cdot \delta \cdot \min(\lambda, \ell),$$

which concludes the proof.

(b). As $\ell = 1$, $\mathrm{H}_{\mathrm{ber}}(\mathbf{q}, 1) = 1 - \prod_{i=1}^{n} (1 - q_i) \stackrel{\text{def}}{=} 1 - e^{-s}$, where $s = -\sum_{i=1}^{n} \ln(1 - q_i)$. Let $\lambda = \sum_{i=1}^{n} q_i$, then $\mathrm{H}_{\mathrm{pois}}(\mathbf{q}, 1) = 1 - e^{-\lambda}$. Observe that $q_i \leq -\ln(1 - q_i) \leq \frac{q_i}{1 - q_i} \leq \frac{q_i}{1 - \delta}$. Thus $\lambda \leq s \leq \frac{\lambda}{1 - \delta}$. The former inequality implies the desired lower bound $\mathrm{H}_{\mathrm{ber}}(\mathbf{q}, 1) - \mathrm{H}_{\mathrm{pois}}(\mathbf{q}, 1) \geq 0$.

To prove the required upper bound, observe that $\mathrm{H}_{\mathrm{ber}}(\mathbf{q}, 1) = f(s)$, $\mathrm{H}_{\mathrm{pois}}(\mathbf{q}, 1) = f(\lambda)$ for $f(t) = 1 - e^{-t}$. As $f(t)$ is a concave function with $f(0) = 0$, we have $f(\lambda) \geq f(s) \cdot \frac{\lambda}{s} \geq f(s) \cdot (1 - \delta)$. Thus $\mathrm{H}_{\mathrm{ber}}(\mathbf{q}, 1) - \mathrm{H}_{\mathrm{pois}}(\mathbf{q}, 1) = f(s) - f(\lambda) \leq \delta \cdot f(s) = \delta \cdot \mathrm{H}_{\mathrm{ber}}(\mathbf{q}, 1)$, which concludes the proof.  □

*Algorithm for Small Tail Probabilities.* Assume that each bidder's value $v_i$ has at most $\delta$ probability to be greater than zero. Then Lemma A.2 and Claim 4.2 (Concavity of Poisson) suggest Algorithm 3.

---

**Algorithm 3: Approximation to Fractional BSP for Tail Probabilities $\ell$-unit Auctions.**

Return the optimal solution $\widetilde{\mathbf{x}}^*$ to the concave program:

$$\begin{aligned} \text{Maximize} \quad & \widetilde{\mathrm{SW}}(\mathbf{x}, \ell) \stackrel{\text{def}}{=} \int_0^{+\infty} \mathrm{H}_{\mathrm{pois}}(\mathbf{q}(\mathbf{x}, \tau), \ell)\, \mathrm{d}\tau \\ \text{Subject To} \quad & \sum_{i=1}^{n} x_i \leq k, \quad x_i \in [0, 1], \forall i \in \{1, 2, \ldots, n\}. \end{aligned}$$

---

We can solve the above program efficiently via standard concave function maximization methods, as the objective $\widetilde{\mathrm{SW}}$ is concave in $\mathbf{x}$. Indeed, from Claim 4.2 we know that $\mathrm{H}_{\mathrm{pois}}(\mathbf{q}, \ell)$ is concave in $\mathbf{q}$ and, since $\mathbf{q}(\mathbf{x}, \tau)$ is linear in $\mathbf{x}$ for every fixed $\tau$, $\mathrm{H}_{\mathrm{pois}}$ is also concave in $\mathbf{x}$. As $\widetilde{\mathrm{SW}}(\mathbf{x}, \ell)$ is an integral (non-negative linear combination) of $\mathrm{H}_{\mathrm{pois}}(\mathbf{q}(\mathbf{x}, \tau), \ell)$, $\widetilde{\mathrm{SW}}$ is concave in $\mathbf{x}$.

From Lemma A.2, we obtain the following approximation guarantees of $\mathrm{SW}(\mathbf{x}, \ell)$ by $\widetilde{\mathrm{SW}}(\mathbf{x}, \ell)$.

$$\left| \mathrm{SW}(\mathbf{x}, \ell) - \widetilde{\mathrm{SW}}(\mathbf{x}, \ell) \right| \leq \int_0^{+\infty} \left| \mathrm{H}_{\mathrm{ber}}(\mathbf{q}(\mathbf{x}, \tau), \ell) - \mathrm{H}_{\mathrm{pois}}(\mathbf{q}(\mathbf{x}, \tau), \ell) \right| \mathrm{d}\tau$$

$$\leq 17.5\, \delta \cdot \mathrm{SW}(\mathbf{x}, \ell). \tag{11}$$

Let $\mathbf{x}^*$ be the best solution of the original problem (1). We have

$$\mathrm{SW}(\widetilde{\mathbf{x}}^*, \ell) \geq \widetilde{\mathrm{SW}}(\widetilde{\mathbf{x}}^*, \ell) - 17.5\, \delta \cdot \mathrm{SW}(\widetilde{\mathbf{x}}^*, \ell)$$

$$\geq \widetilde{\mathrm{SW}}(\mathbf{x}^*, \ell) - 17.5\, \delta \cdot \mathrm{SW}(\mathbf{x}^*, \ell) \geq (1 - 35\, \delta) \mathrm{SW}(\mathbf{x}^*, \ell).$$

Hence, Algorithm 3 is $(1 - 35\, \delta)$-approximation.

*Chernoff Approximation Guarantees.* We recall the following definitions

$$\mathrm{H}_{\mathrm{ber}}(\mathbf{q}, \ell) = \mathop{\mathbf{E}}_{\mathbf{z} \sim \mathrm{Ber}(\mathbf{q})} \left[ \min \left( \sum_{i=1}^{n} z_i, \ell \right) \right], \qquad \mathrm{H}_{\mathrm{cher}}(\mathbf{q}, \ell) = \min \left( \sum_{i=1}^{n} q_i, \ell \right).$$

Our main algorithm needs the following approximation guarantee of $\mathrm{H}_{\mathrm{cher}}$:

**LEMMA A.3.** *For all* $\mathbf{q} \in [0,1]^n$, $\ell \in \mathbb{N}^+$, *let* $\lambda = \sum_{i=1}^n q_i$, *the following properties hold.*

(a)
$$\mathrm{H}_{\mathrm{ber}}(\mathbf{q},\ell) \leq \mathrm{H}_{\mathrm{cher}}(\mathbf{q},\ell) \leq 7 \cdot \mathrm{H}_{\mathrm{ber}}(\mathbf{q},\ell).$$

(b)
$$\mathrm{H}_{\mathrm{cher}}(\mathbf{q},\ell) - \mathrm{H}_{\mathrm{ber}}(\mathbf{q},\ell) \leq \frac{3}{\sqrt{\lambda}} \cdot \mathrm{H}_{\mathrm{cher}}(\mathbf{q},\ell) \leq \frac{21}{\sqrt{\lambda}} \cdot \mathrm{H}_{\mathrm{ber}}(\mathbf{q},\ell).$$

(c)
$$\mathrm{H}_{\mathrm{cher}}(\mathbf{q},\ell) - \mathrm{H}_{\mathrm{ber}}(\mathbf{q},\ell) \leq \frac{5}{\sqrt{\ell}} \cdot \mathrm{H}_{\mathrm{cher}}(\mathbf{q},\ell) \leq \frac{35}{\sqrt{\ell}} \cdot \mathrm{H}_{\mathrm{ber}}(\mathbf{q},\ell).$$

In order to prove Lemma A.3, we first prove the following two auxiliary lemmas.

**LEMMA A.4.** *For all* $\mathbf{q} \in [0,1]^n$, $\ell \in \mathbb{N}^+$, *let* $\lambda = \sum_{i=1}^n q_i$, *the following properties hold.*

(a) $\mathrm{H}_{\mathrm{ber}}(\mathbf{q},\ell) \geq \lambda(1 - \frac{1}{2}\lambda)$.

(b) *If* $\lambda \geq 1$, $\mathrm{H}_{\mathrm{ber}}(\mathbf{q},\ell) \geq \frac{1}{2}$.

(c) *If* $\ell \geq \lambda$, *then* $\mathrm{H}_{\mathrm{cher}}(\mathbf{q},\ell) - \mathrm{H}_{\mathrm{ber}}(\mathbf{q},\ell) \leq \sum_{i=\ell+1}^{n} e^{-\frac{\delta^2(i)}{2+\delta(i)} \cdot \lambda}$, *where* $\delta(i) = \frac{i-\lambda}{\lambda}$.

(d) *If* $\ell \geq \lambda$ *and* $\alpha \in (0,1)$, *then* $\mathrm{H}_{\mathrm{cher}}(\mathbf{q},\ell) - \mathrm{H}_{\mathrm{ber}}(\mathbf{q},\ell) \leq \alpha\lambda + \frac{6}{\alpha} e^{-\alpha^2 \cdot \frac{\lambda}{3}}$.

(e) *If* $\lambda \geq \ell$, *then* $\mathrm{H}_{\mathrm{cher}}(\mathbf{q},\ell) - \mathrm{H}_{\mathrm{ber}}(\mathbf{q},\ell) \leq \sum_{i=0}^{\ell-1} e^{-\frac{(\lambda-i)^2}{2\lambda}}$.

(f) *If* $\lambda \geq \ell$ *and* $\alpha \in (0,1)$, *then* $\mathrm{H}_{\mathrm{cher}}(\mathbf{q},\ell) - \mathrm{H}_{\mathrm{ber}}(\mathbf{q},\ell) \leq \alpha\lambda + \frac{4}{\alpha} e^{-\alpha^2 \cdot \frac{\lambda}{2}}$.

**PROOF OF LEMMA A.4.** (a). Since $\ell \geq 1$, We have the following lower bound on $\mathrm{H}_{\mathrm{ber}}(\mathbf{q},\ell)$

$$\mathrm{H}_{\mathrm{ber}}(\mathbf{q},\ell) \geq \Pr_{\substack{\mathbf{z}\sim\mathrm{Ber}(\mathbf{q}) \\ Z=\sum_{i=1}^n z_i}}[Z \geq 1] = \sum_{i=1}^n \Pr_{z_i\sim\mathrm{Ber}(q_i)}[z_i = 1] \cdot \Pr_{\mathbf{z}\sim\mathrm{Ber}(\mathbf{q})}\left[\forall j < i, \ z_j = 0\right]$$

$$\geq \sum_{i=1}^n q_i \cdot \left(1 - \sum_{j=1}^{i-1} q_j\right) = \sum_{i=1}^n q_i - \sum_{j<i} q_i q_j \geq \lambda\left(1 - \frac{1}{2}\lambda\right).$$

(b). If $\lambda \geq 1$, there exists $\mathbf{q}' \in [0,1]^n$ such that $q'_i \leq q_i$, $\forall i \in \{1,2,\ldots,n\}$ and $\sum_{i=1}^n q'_i = 1$. Using Lemma A.4 (a), we can see that $\mathrm{H}_{\mathrm{ber}}(\mathbf{q}',\ell) \geq 0.5$. Then $\mathrm{H}_{\mathrm{ber}}(\mathbf{q},\ell) \geq \mathrm{H}_{\mathrm{ber}}(\mathbf{q}',\ell) \geq 0.5$.

(c). As $\ell \geq \lambda$, we have $\mathrm{H}_{\mathrm{cher}}(\mathbf{q},\ell) = \lambda$. Then

$$\mathrm{H}_{\mathrm{cher}}(\mathbf{q},\ell) - \mathrm{H}_{\mathrm{ber}}(\mathbf{q},\ell) = \mathop{\mathbb{E}}_{\substack{\mathbf{z}\sim\mathrm{Ber}(\mathbf{q}) \\ Z=\sum_{i=1}^n z_i}}[Z - \min\{Z,\ell\}] = \sum_{i=\ell+1}^{n} \Pr_{\substack{\mathbf{z}\sim\mathrm{Ber}(\mathbf{q}) \\ Z=\sum_{i=1}^n z_i}}[Z \geq i].$$

We apply Chernoff bound to each tail probability $Z \geq i$ under the sum. Thus

$$\mathrm{H}_{\mathrm{cher}}(\mathbf{q},\ell) - \mathrm{H}_{\mathrm{ber}}(\mathbf{q},\ell) \leq \sum_{i=\ell+1}^{n} e^{-\frac{\delta^2(i)}{2+\delta(i)} \cdot \lambda}, \quad \text{where } \delta(i) = \frac{i-\lambda}{\lambda}$$

(d). By Lemma A.4 (c)

$$\mathrm{H}_{\mathrm{cher}}(\mathbf{q},\ell) - \mathrm{H}_{\mathrm{ber}}(\mathbf{q},\ell) \leq \sum_{i=\lceil\lambda+1\rceil}^{\infty} e^{-\frac{\delta^2(i)}{2+\delta(i)} \cdot \lambda} \leq \alpha\lambda + \sum_{i=\lceil(1+\alpha)\lambda\rceil}^{\infty} e^{-\frac{\delta^2(i)}{2+\delta(i)} \cdot \lambda}.$$

Note that inside the last summation, $\delta(i) \geq \alpha$ and $\frac{\delta(i)}{2+\delta(i)} \geq \frac{\alpha}{2+\alpha} \geq \frac{\alpha}{3}$. Therefore,

$$\mathrm{H}_{\mathrm{cher}}(\mathbf{q},\ell) - \mathrm{H}_{\mathrm{ber}}(\mathbf{q},\ell) \leq \alpha\lambda + \sum_{i=\lceil(1+\alpha)\lambda\rceil}^{\infty} e^{-\frac{\alpha}{2+\alpha} \cdot \delta(i)\lambda} \leq \alpha\lambda + \sum_{i=\lceil(1+\alpha)\lambda\rceil}^{\infty} e^{-(i-\lambda)\cdot\frac{\alpha}{3}}.$$

As the summation in the right hand side becomes a geometric series, we get

$$\mathrm{H}_{\mathrm{cher}}(\mathbf{q},\ell) - \mathrm{H}_{\mathrm{ber}}(\mathbf{q},\ell) \leq \alpha\lambda + e^{-\alpha^2\cdot\frac{\lambda}{3}} \cdot \left(1 - e^{-\frac{\alpha}{3}}\right)^{-1} \leq \alpha\lambda + \frac{6}{\alpha} e^{-\alpha^2\cdot\frac{\lambda}{3}}.$$

The last inequality holds as $1 - e^{-x} \geq x/2$ for $x \in [0,1]$. Hence, Lemma A.4 (d) holds.

(e). If $\ell \leq \lambda$, then $\mathrm{H}_{\mathrm{cher}}(\mathbf{q},\ell) = \ell$. We have

$$\mathrm{H}_{\mathrm{cher}}(\mathbf{q},\ell) - \mathrm{H}_{\mathrm{ber}}(\mathbf{q},\ell) = \mathop{\mathbb{E}}_{\substack{\mathbf{z}\sim\mathrm{Ber}(\mathbf{q}) \\ Z=\sum_{i=1}^n z_i}}[\ell - \min\{Z,\ell\}] = \sum_{i=0}^{\ell-1} \Pr_{\substack{\mathbf{z}\sim\mathrm{Ber}(\mathbf{q}) \\ Z=\sum_{i=1}^n z_i}}[Z \leq i].$$

We again apply Chernoff bound for each tail event $Z \leq i$ under summation and get

$$H_{\text{cher}}(\mathbf{q}, \ell) - H_{\text{ber}}(\mathbf{q}, \ell) \leq \sum_{i=0}^{\ell-1} e^{-\frac{(\lambda-i)^2}{2\lambda}}.$$

(f). By Lemma A.4 (e), we have

$$H_{\text{cher}}(\mathbf{q}, \ell) - H_{\text{ber}}(\mathbf{q}, \ell) \leq \sum_{i=0}^{\ell-1} e^{-\frac{(\lambda-i)^2}{2\lambda}} \leq \sum_{i=1}^{\infty} e^{-\frac{i^2}{2\lambda}} \leq \alpha\lambda + \sum_{i=\lceil\alpha\lambda\rceil}^{\infty} e^{-\frac{i^2}{2\lambda}} \leq \alpha\lambda + \sum_{i=\lceil\alpha\lambda\rceil}^{\infty} e^{-i\cdot\frac{\alpha}{2}}.$$

The summation in the right hand side is again a geometric series, which allows us to get

$$H_{\text{cher}}(\mathbf{q}, \ell) - H_{\text{ber}}(\mathbf{q}, \ell) \leq \alpha\lambda + e^{-\alpha^2\cdot\frac{\lambda}{2}} \cdot (1 - e^{-\frac{\alpha}{2}})^{-1} \leq \alpha\lambda + \frac{4}{\alpha}e^{-\alpha^2\cdot\frac{\lambda}{2}}.$$

Hence Lemma A.4 (f) holds. □

CLAIM A.5. *For any* $\alpha \geq 3.4$, $\sum_{i=1}^{\infty} e^{-\frac{i^2}{2\alpha+i}} \leq 0.85\alpha$.

PROOF. When $\alpha, x > 0$, the function $-\frac{x^2}{2\alpha+x}$ is decreasing in $x$. Therefore,

$$\sum_{x=1}^{\infty} e^{-\frac{x^2}{2\alpha+x}} \leq \int_{y=0}^{+\infty} e^{-\frac{y^2}{2\alpha+y}} \, dy \overset{[y=\alpha\cdot z]}{=} \alpha \cdot \int_{z=0}^{+\infty} e^{-\frac{\alpha z^2}{2+z}} \, dz \leq \alpha \cdot \int_{z=0}^{+\infty} e^{-\frac{3.4z^2}{2+z}} \, dz \leq \alpha \cdot 0.85. \qquad \square$$

Now we are ready to prove Lemma A.3.

LEMMA A.3. *For all* $\mathbf{q} \in [0,1]^n, \ell \in \mathbb{N}^+$, *let* $\lambda = \sum_{i=1}^{n} q_i$, *the following properties hold.*

(a) $$H_{\text{ber}}(\mathbf{q}, \ell) \leq H_{\text{cher}}(\mathbf{q}, \ell) \leq 7 \cdot H_{\text{ber}}(\mathbf{q}, \ell).$$

(b) $$H_{\text{cher}}(\mathbf{q}, \ell) - H_{\text{ber}}(\mathbf{q}, \ell) \leq \frac{3}{\sqrt{\lambda}} \cdot H_{\text{cher}}(\mathbf{q}, \ell) \leq \frac{21}{\sqrt{\lambda}} \cdot H_{\text{ber}}(\mathbf{q}, \ell).$$

(c) $$H_{\text{cher}}(\mathbf{q}, \ell) - H_{\text{ber}}(\mathbf{q}, \ell) \leq \frac{5}{\sqrt{\ell}} \cdot H_{\text{cher}}(\mathbf{q}, \ell) \leq \frac{35}{\sqrt{\ell}} \cdot H_{\text{ber}}(\mathbf{q}, \ell).$$

PROOF OF LEMMA A.3. (a). As $\min\{t, \ell\}$ is a concave function in $t$, the Jensen inequality gives us

$$H_{\text{cher}}(\mathbf{q}, \ell) = \min\left\{\mathop{\mathbf{E}}_{\substack{\mathbf{z}\sim\text{Ber}(\mathbf{q})\\ Z=\sum_{i=1}^{n} z_i}}[Z], \ell\right\} \geq \mathop{\mathbf{E}}_{\substack{\mathbf{z}\sim\text{Ber}(\mathbf{q})\\ Z=\sum_{i=1}^{n} z_i}}[\min\{Z, \ell\}] = H_{\text{ber}}(\mathbf{q}, \ell),$$

which gives the first inequality. To prove the second inequality we consider the following 4 cases:

- If $\lambda \leq 1$, we use Lemma A.4 (a) to get $H_{\text{ber}}(\mathbf{q}, \ell) \geq \lambda\left(1 - \frac{1}{2}\lambda\right) \geq \frac{1}{2}\lambda \geq \frac{1}{2} \cdot H_{\text{cher}}(\mathbf{q}, \ell)$.
- If $\lambda > 1$ and $\min(\lambda, \ell) \leq 3.5$, then Lemma A.4 (b) gives us $H_{\text{ber}}(\mathbf{q}, \ell) \geq \frac{1}{2} \geq \frac{1}{7} \cdot \min(\lambda, \ell) = \frac{1}{7} \cdot H_{\text{cher}}(\mathbf{q}, \ell)$.
- If $\min(\lambda, \ell) > 3.5$ and $\ell \geq \lambda$, then Lemma A.4 (c) gives us

$$H_{\text{cher}}(\mathbf{q}, \ell) - H_{\text{ber}}(\mathbf{q}, \ell) \leq \sum_{i=\ell+1}^{n} e^{-\frac{\delta^2(i)}{2+\delta(i)}\cdot\lambda} \leq \sum_{i=\lceil\lambda+1\rceil}^{\infty} e^{-\frac{(i-\lambda)^2}{2\lambda+(i-\lambda)}} \leq \sum_{i=1}^{\infty} e^{-\frac{i^2}{2\lambda+i}}.$$

  We apply Claim A.5 for $\alpha = \lambda$ and get $H_{\text{cher}}(\mathbf{q}, \ell) - H_{\text{ber}}(\mathbf{q}, \ell) \leq 0.85\lambda$, which implies that $H_{\text{ber}}(\mathbf{q}, \ell) \geq 0.15 \cdot H_{\text{cher}}(\mathbf{q}, \ell)$ for $H_{\text{cher}}(\mathbf{q}, \ell) = \lambda \leq \ell$.

- If $\min(\lambda, \ell) > 3.5$ and $\lambda \geq \ell$, then Lemma A.4 (e) gives us $H_{\text{cher}}(\mathbf{q}, \ell) - H_{\text{ber}}(\mathbf{q}, \ell) \leq \sum_{i=0}^{\ell-1} e^{-\frac{1}{2\lambda}(\lambda-i)^2}$. Note that $\frac{1}{2\lambda}(\lambda-i)^2 \geq \frac{1}{2\ell}(\ell-i)^2$, when $\lambda \geq \ell \geq i$. Thus

$$H_{\text{cher}}(\mathbf{q}, \ell) - H_{\text{ber}}(\mathbf{q}, \ell) \leq \sum_{i=0}^{\ell-1} e^{-\frac{1}{2\ell}(\ell-i)^2} \leq \sum_{i=1}^{\infty} e^{-\frac{i^2}{2\ell}} \leq \sum_{i=1}^{\infty} e^{-\frac{i^2}{2\ell+i}}.$$

  By applying Claim A.5 with $\alpha = \ell$, we get $H_{\text{cher}}(\mathbf{q}, \ell) - H_{\text{ber}}(\mathbf{q}, \ell) \leq 0.85\ell$, which implies that $H_{\text{ber}}(\mathbf{q}, \ell) \geq 0.15 \cdot H_{\text{cher}}(\mathbf{q}, \ell)$, as $H_{\text{cher}}(\mathbf{q}, \ell) = \ell \leq \lambda$.

In all 4 cases, $7H_{\text{ber}}(\mathbf{q}, \ell) \geq H_{\text{cher}}(\mathbf{q}, \ell)$.

(b). Let $\varepsilon \overset{\text{def}}{=} \frac{1}{\sqrt{\lambda}}$. We consider three cases to show that $H_{\text{cher}}(\mathbf{q}, \ell) - H_{\text{ber}}(\mathbf{q}, \ell) \leq 3\varepsilon \cdot H_{\text{cher}}(\mathbf{q}, \ell)$, then combine it with the inequality $7H_{\text{ber}}(\mathbf{q}, \ell) \geq H_{\text{cher}}(\mathbf{q}, \ell)$ from Lemma A.3 (a) to conclude the proof. Without loss of generality, we may assume that $\varepsilon < 1/3$.

- If $\ell \geq \lambda$, then Lemma A.4 (d) for $\alpha = 1.8\varepsilon$, gives us $H_{\text{cher}}(\mathbf{q}, \ell) - H_{\text{ber}}(\mathbf{q}, \ell) \leq 2.94\varepsilon\lambda = 2.94\varepsilon \cdot H_{\text{cher}}(\mathbf{q}, \ell)$.

- If $\ell \leq \frac{3}{4}\lambda$, then by Lemma A.4 (e) we have

$$\mathrm{H}_{\mathrm{cher}}(\mathbf{q}, \ell) - \mathrm{H}_{\mathrm{ber}}(\mathbf{q}, \ell) \leq \sum_{i=0}^{\ell-1} e^{-\frac{(\lambda-i)^2}{2\lambda}} \leq \ell \cdot e^{-\frac{\lambda}{32}} \leq \frac{4e^{-1/2}}{\sqrt{\lambda}} \cdot \mathrm{H}_{\mathrm{cher}}(\mathbf{q}, \ell) \leq 2.5 \cdot \varepsilon \cdot \mathrm{H}_{\mathrm{cher}}(\mathbf{q}, \ell),$$

  where the first inequality holds because $\lambda - i \geq \lambda/4$; the second inequality holds, as $\mathrm{H}_{\mathrm{cher}}(\mathbf{q}, \ell) = \ell$ and $e^{-x/c}\sqrt{x} \leq e^{-1/2}\sqrt{c/2}$ for all $x, c > 0$.
- If $\ell \in \left(\frac{3}{4}\lambda, \lambda\right)$, by Lemma A.4 (f) for $\alpha = 1.8\varepsilon$ we have

$$\mathrm{H}_{\mathrm{cher}}(\mathbf{q}, \ell) - \mathrm{H}_{\mathrm{ber}}(\mathbf{q}, \ell) \leq 2.24\varepsilon\lambda \leq 2.99\varepsilon\ell = 2.99\varepsilon \cdot \mathrm{H}_{\mathrm{cher}}(\mathbf{q}, \ell).$$

(c). Let $\varepsilon \stackrel{\mathrm{def}}{=} \frac{1}{\sqrt{\ell}}$. We first prove $\mathrm{H}_{\mathrm{cher}}(\mathbf{q}, \ell) - \mathrm{H}_{\mathrm{ber}}(\mathbf{q}, \ell) \leq 5\varepsilon \cdot \mathrm{H}_{\mathrm{cher}}(\mathbf{q}, \ell)$ by considering the following four cases. It implies the second bound when combined with inequality $7\mathrm{H}_{\mathrm{ber}}(\mathbf{q}, \ell) \geq \mathrm{H}_{\mathrm{cher}}(\mathbf{q}, \ell)$ from Lemma A.3 (a). We may assume without loss of generality that $\varepsilon < 1/5$.

- If $\lambda < \frac{1}{\sqrt{\ell}}$, then $\mathrm{H}_{\mathrm{cher}}(\mathbf{q}, \ell) = \lambda$ and Lemma A.4 (a) gives us

$$\mathrm{H}_{\mathrm{cher}}(\mathbf{q}, \ell) - \mathrm{H}_{\mathrm{ber}}(\mathbf{q}, \ell) \leq \lambda - \lambda\left(1 - \frac{1}{2}\lambda\right) = \frac{\lambda^2}{2} < \frac{\varepsilon\lambda}{2} = \frac{\varepsilon}{2} \cdot \mathrm{H}_{\mathrm{cher}}(\mathbf{q}, \ell).$$

- If $\lambda \in [\frac{1}{\sqrt{\ell}}, \frac{\ell}{3})$, then $\mathrm{H}_{\mathrm{cher}}(\mathbf{q}, \ell) = \lambda$ and by Lemma A.4 (c) we have

$$\mathrm{H}_{\mathrm{cher}}(\mathbf{q}, \ell) - \mathrm{H}_{\mathrm{ber}}(\mathbf{q}, \ell) \leq \sum_{i=\ell+1}^{n} e^{-\frac{\delta^2(i)}{2+\delta(i)}\cdot\lambda} \leq \sum_{i=\ell+1}^{n} e^{-\delta(i)\cdot\frac{\lambda}{2}} \leq \sum_{i=\ell+1}^{+\infty} e^{-\frac{i-\lambda}{2}},$$

  where the second inequality holds, as $\delta(i) = \frac{i-\lambda}{\lambda} \geq 2$ for all $i \geq \ell + 1$ and $\lambda < \ell/3$. Furthermore,

$$\mathrm{H}_{\mathrm{cher}}(\mathbf{q}, \ell) - \mathrm{H}_{\mathrm{ber}}(\mathbf{q}, \ell) \leq e^{-\frac{\ell}{3}} \cdot (1 - e^{-\frac{1}{2}})^{-1} \leq \frac{3e^{-1}/\ell}{1 - e^{-1/2}} \leq 2.81 \cdot \varepsilon \cdot \mathrm{H}_{\mathrm{cher}}(\mathbf{q}, \ell),$$

  where to get the first inequality we simply use the formula for the sum of geometric progression and estimate $e^{-\frac{\ell+1-\lambda}{2}} \leq e^{-\ell/3}$; the second inequality holds, because $e^{-x/c} \leq c \cdot e^{-1}/x$ for any $x, c > 0$; the last inequality holds, because $\mathrm{H}_{\mathrm{cher}}(\mathbf{q}, \ell) = \lambda \geq \varepsilon = 1/\sqrt{\ell}$.
- If $\lambda \in [\frac{\ell}{3}, \ell]$, then $\mathrm{H}_{\mathrm{cher}}(\mathbf{q}, \ell) = \lambda$ and by Lemma A.4 (d) for $\alpha = 4\varepsilon$ ($\alpha < 1$, since $\varepsilon \leq 1/5$) we get $\mathrm{H}_{\mathrm{cher}}(\mathbf{q}, \ell) - \mathrm{H}_{\mathrm{ber}}(\mathbf{q}, \ell) \leq 4\varepsilon\lambda + \frac{6}{4\varepsilon}e^{-16\varepsilon^2\cdot\frac{\lambda}{3}} \leq 4.77\varepsilon\lambda = 4.77\varepsilon \cdot \mathrm{H}_{\mathrm{cher}}(\mathbf{q}, \ell)$, where the second inequality holds, since $\ell \leq 3\lambda$ and $\frac{6}{4\varepsilon}e^{-16\varepsilon^2\cdot\frac{\lambda}{3}} = \frac{3\varepsilon\ell}{2}e^{-16\cdot\frac{\lambda}{3\ell}} \leq \frac{9\varepsilon\lambda}{2}e^{-\frac{16\ell}{9\ell}} \leq 0.77 \cdot \varepsilon\lambda$.
- If $\lambda > \ell$, then by considering $\mathbf{q}'$ with $q_i' \leq q_i$ and $\sum_{i=1}^{n} q_i' = \ell$, we get $\mathrm{H}_{\mathrm{ber}}(\mathbf{q}) \geq \mathrm{H}_{\mathrm{ber}}(\mathbf{q}')$ and $\mathrm{H}_{\mathrm{cher}}(\mathbf{q}) = \mathrm{H}_{\mathrm{cher}}(\mathbf{q}') = \ell$. Then according to the previous case, $\mathrm{H}_{\mathrm{cher}}(\mathbf{q}, \ell) - \mathrm{H}_{\mathrm{ber}}(\mathbf{q}, \ell) \leq \mathrm{H}_{\mathrm{cher}}(\mathbf{q}', \ell) - \mathrm{H}_{\mathrm{ber}}(\mathbf{q}', \ell) \leq 4.77\varepsilon \cdot \mathrm{H}_{\mathrm{cher}}(\mathbf{q}', \ell) = 4.77\varepsilon \cdot \mathrm{H}_{\mathrm{cher}}(\mathbf{q}, \ell)$.

These bounds combined with inequality $7\mathrm{H}_{\mathrm{ber}}(\mathbf{q}, \ell) \geq \mathrm{H}_{\mathrm{cher}}(\mathbf{q}, \ell)$ conclude the proof of (c). □

# B  MISSING PROOFS

## B.1  Proof of Theorem 4.4

THEOREM 4.4. *Algorithm 1 works in polynomial time and is a $(1 - O(\varepsilon))$-approximation, i.e., $(1 - 43\,k^{-1/4})$-approximation to the fractional BSP for any position auction.*

PROOF OF THEOREM 4.4. We first show that Algorithm 1 is polynomial. Indeed, step (1) works in polynomial time by Claim 4.3. For each $\tau$ in the support of $\{D_i\}_{i \in [n]}$ and $\mathbf{x}_M$ we can efficiently calculate $\mathrm{H}_{\mathrm{cher}}((\mathbf{x}_M, \mathbf{1}_{S_{\mathrm{fix}}}), \tau)$ and $\mathrm{G}_{\mathrm{pois}}(\mathbf{x}_M, \tau)$, which allows us to compute $\widetilde{\mathrm{SW}}(\mathbf{x}_M)$ in polynomial time. It is easy to see that $\widetilde{\mathrm{SW}}(\mathbf{x}_M)$ is a concave function in $\mathbf{x}_M$, as $\mathrm{G}_{\mathrm{pois}}$ is a non-negative linear combination of constant terms and concave functions $\mathrm{H}_{\mathrm{pois}}(\mathbf{x}_M, \ell-j, \tau)$, and $\mathrm{H}_{\mathrm{cher}}$ is a non-negative linear combination of concave functions $\mathrm{H}_{\mathrm{cher}}(\mathbf{q}(\mathbf{x}), \tau), \ell)$ in $\mathbf{x}$. Furthermore, given the representation of $\widetilde{\mathrm{SW}}(\mathbf{x}_M)$ as an integral of nice algebraic functions $\mathrm{H}_{\mathrm{cher}}$ and $\mathrm{G}_{\mathrm{pois}}$, we can also compute all first and second order derivatives of $\widetilde{\mathrm{SW}}(\mathbf{x}_M)$ in polynomial time. This allows us to find the optimal solution $\widetilde{\mathbf{x}}_M^*$ to (9) in polynomial time using standard concave (first or second order) maximization methods.

To get the stated approximation guarantee we first compare the original objective $\mathrm{SW}(\mathbf{x}_M, \mathbf{1}_{S_{\mathrm{fix}}})$ in (4) with $\widetilde{\mathrm{SW}}(\mathbf{x}_M)$ and get the following Lemma.

LEMMA B.1. *For any $\mathbf{x} = (\mathbf{x}_M, \mathbf{1}_{S_{\mathrm{fix}}})$ with $\mathbf{x}_M \in [0, 1]^{|M|}$ and any weights vector $\mathbf{w}$,*

$$\left|\widetilde{\mathrm{SW}}(\mathbf{x}_M) - \mathrm{SW}(\mathbf{x})\right| \leq 21\,\varepsilon \cdot \mathrm{SW}(\mathbf{x}).$$

**Proof.** We rewrite $\mathrm{SW}(\mathbf{x}, \mathbf{w})$ for $\mathbf{x} = (\mathbf{x}_M, \mathbf{1}_{S_{\mathrm{fix}}})$ in the same form as (8).

$$\mathrm{SW}(\mathbf{x}) = \int_0^\eta \mathrm{H}_{\mathrm{ber}}(\mathbf{x}, \tau)\, \mathrm{d}\tau + \int_\eta^{+\infty} \mathrm{G}_{\mathrm{ber}}(\mathbf{x}_M, \tau)\, \mathrm{d}\tau,$$

where $\quad \mathrm{G}_{\mathrm{ber}}(\mathbf{x}_M, \tau) \overset{\mathrm{def}}{=} \sum_{j=0}^{|S_{\mathrm{fix}}|} \mathbf{Pr}[Z_{\mathrm{fix}}(\tau) = j] \cdot \left( \sum_{\ell=1}^{j} w_\ell + \sum_{\ell=j+1}^{n} (w_\ell - w_{\ell+1}) \cdot \mathrm{H}_{\mathrm{ber}}(\mathbf{x}_M, \ell-j, \tau) \right).$

We first compare the corresponding adjusted Bernoulli and Poisson terms. Observe that $q_i(\mathbf{x}, \tau) \le \delta = \varepsilon$ for each bidder $i \in M$ and threshold $\tau > \eta$ according to condition (b) in Claim 4.3. Thus by Lemma A.2 (a),

$$\left| \mathrm{G}_{\mathrm{pois}}(\mathbf{x}_M, \tau) - \mathrm{G}_{\mathrm{ber}}(\mathbf{x}_M, \tau) \right|$$

$$\le \sum_{j=0}^{|S_{\mathrm{fix}}|} \mathbf{Pr}[Z_{\mathrm{fix}}(\tau) = j] \cdot \sum_{\ell=j+1}^{n} (w_\ell - w_{\ell+1}) \cdot \left| \mathrm{H}_{\mathrm{pois}}(\mathbf{x}_M, \ell-j, \tau) - \mathrm{H}_{\mathrm{ber}}(\mathbf{x}_M, \ell-j, \tau) \right|$$

$$\le \sum_{j=0}^{|S_{\mathrm{fix}}|} \mathbf{Pr}[Z_{\mathrm{fix}}(\tau) = j] \cdot \sum_{\ell=j+1}^{n} (w_\ell - w_{\ell+1}) \cdot 17.5\, \delta \cdot \mathrm{H}_{\mathrm{ber}}(\mathbf{x}_M, \ell-j, \tau)$$

$$\le 17.5\, \varepsilon \cdot \mathrm{G}_{\mathrm{ber}}(\mathbf{x}_M, \tau).$$

Next, we compare Chernoff and Bernoulli objectives ($\mathrm{H}_{\mathrm{cher}}$ and $\mathrm{H}_{\mathrm{ber}}$) for low range thresholds $\tau \le \eta$. As both $\mathrm{H}_{\mathrm{cher}}(\mathbf{q}, \mathbf{w})$ and $\mathrm{H}_{\mathrm{ber}}(\mathbf{q}, \mathbf{w})$ are non-negative linear combinations of respective terms for $\ell$-unit auctions with $\sum_{i \in [n]} q_i(x_i, \tau) \ge \ell^*$ for $\tau \le \eta$, we get by Lemma A.3 (b) that

$$|\mathrm{H}_{\mathrm{cher}}(\mathbf{q}, \mathbf{w}) - \mathrm{H}_{\mathrm{ber}}(\mathbf{q}, \mathbf{w})| \le 21 \cdot \frac{\mathrm{H}_{\mathrm{ber}}(\mathbf{q}, \mathbf{w})}{\sqrt{\ell^*}} \le 21\, \varepsilon \cdot \mathrm{H}_{\mathrm{ber}}(\mathbf{q}, \mathbf{w}).$$

Thus after combining the two bounds for high and low ranges of thresholds $\tau$ we get

$$\left| \widetilde{\mathrm{SW}}(\mathbf{x}_M) - \mathrm{SW}(\mathbf{x}) \right| \le 21\, \varepsilon \cdot \int_0^\eta \mathrm{H}_{\mathrm{ber}}(\mathbf{x}, \tau)\, \mathrm{d}\tau + 17.5\, \varepsilon \cdot \int_\eta^{+\infty} \mathrm{G}_{\mathrm{ber}}(\mathbf{x}_M, \tau)\, \mathrm{d}\tau \le 21\, \varepsilon \cdot \mathrm{SW}(\mathbf{x}),$$

which concludes the proof of Lemma B.1. $\qquad\square$

We proceed the proof of Theorem 4.4 by letting $\mathbf{x}^*$ be the optimal solution to fractional BSP in (4). Then, we consider $\mathbf{x}_+^* \in \mathbb{R}_{\ge 0}^n$ defined as $\mathbf{x}_+^* \overset{\mathrm{def}}{=} (\mathbf{x}_M^*, \mathbf{1}_{S_{\mathrm{fix}}})$, so that $\mathbf{x}_+^* \ge \mathbf{x}^*$. By Lemma B.1, for $\mathbf{x} = \mathbf{x}_+^*$, we have

$$\widetilde{\mathrm{SW}}(\mathbf{x}_M^*) \ge (1 - 21\, \varepsilon) \cdot \mathrm{SW}(\mathbf{x}_+^*) \ge (1 - 21\, \varepsilon) \cdot \mathrm{SW}(\mathbf{x}^*).$$

On the other hand, by Lemma B.1 for $\mathbf{x} = \widetilde{\mathbf{x}}^*$ we have

$$(1 + 21\, \varepsilon) \cdot \mathrm{SW}(\widetilde{\mathbf{x}}^*) \ge \widetilde{\mathrm{SW}}(\widetilde{\mathbf{x}}_M^*) \ge \widetilde{\mathrm{SW}}\left( \frac{k - \varepsilon \cdot k}{k} \cdot \mathbf{x}_M^* \right) \ge (1 - \varepsilon) \cdot \widetilde{\mathrm{SW}}(\mathbf{x}_M^*) \ge (1 - \varepsilon) \cdot (1 - 21\, \varepsilon) \cdot \mathrm{SW}(\mathbf{x}^*),$$

where the second inequality holds, as $\widetilde{\mathbf{x}}_M^*$ is the optimal solution to (9) and $\frac{k - \varepsilon \cdot k}{k} \cdot \mathbf{x}_M^*$ is a feasible solution; the third inequality holds, as $\widetilde{\mathrm{SW}}(\mathbf{x})$ is a concave function in $\mathbf{x}$; the last inequality holds by the last lower bound on $\widetilde{\mathrm{SW}}(\mathbf{x}_M^*)$. Finally, as $\frac{(1-\varepsilon)(1-21\,\varepsilon)}{1+21\,\varepsilon} \ge 1 - (1 + 2 \cdot 21) \cdot \varepsilon$,

$$\mathrm{SW}(\widetilde{\mathbf{x}}^*) \ge (1 - 43\, \varepsilon) \cdot \mathrm{SW}(\mathbf{x}^*),$$

which concludes the proof of the theorem. $\qquad\square$

## B.2    Proof of Theorem 4.5

**Theorem 4.5.** *Algorithm 2 works in polynomial time and in expectation is a $\left(1 - 43\, k^{-1/4} - O(k^{-1/2})\right)$-approximation to the integral BSP for any position auction.*

**Proof of Theorem 4.5.** Recall that the social welfare $\mathrm{SW}(S)$ of a $\ell$-unit or position auction is submodular as a function of the invited set of bidders $S$.

**Claim B.2.** *The expected social welfare $\mathrm{SW}(S)$ for a set of bidders $S \subseteq [n]$ in any $\ell$-unit or position auction is a submodular function of $S$.*

*Analysis of Algorithm 2.* Clearly, $\mathbf{z}$ is a feasible solution to the integral BSP. We will prove below that Algorithm 2 has almost the same approximation guarantee as Algorithm 1. Let us denote the optimal social welfare for integral BSP as OPT. Then the best solution $\mathbf{x}^*$ to the fractional BSP (4) may have only higher welfare $\mathrm{SW}(\mathbf{x}^*, \mathbf{w}) \geq \mathrm{OPT}$. Furthermore, since (4) is a multi-linear extension of the integral BSP, we have

$$\mathop{\mathbf{E}}_{\mathbf{y} \sim \mathrm{Ber}(\mathbf{x})}[\mathrm{SW}(\mathbf{y}, \mathbf{w})] = \mathrm{SW}(\mathbf{x}, \mathbf{w}) \geq \left(1 - 43 \, k^{-1/4}\right)\mathrm{SW}(\mathbf{x}^*, \mathbf{w}) \geq \left(1 - 43 \, k^{-1/4}\right)\mathrm{OPT}.$$

Our solution $\mathbf{z}$ suffers an additional loss when the sample vector $\mathbf{y}$ has more than $k$ elements $|\mathbf{y}|_1 > k$. On the other hand, for each given $\mathbf{y}$ with $|\mathbf{y}|_1 > k$ we have $\mathbf{E}_{\mathbf{z} \sim \binom{y}{k}}[\mathrm{SW}(\mathbf{z}, \mathbf{w})] \geq \frac{k}{|\mathbf{y}|_1}\mathrm{SW}(\mathbf{y}, \mathbf{w})$, due to submodularity of $\mathrm{SW}(\mathbf{y}, \mathbf{w})$ (here we use a standard fact about monotone non-negative submodular function $f$: a uniformly sampled subset $T \subset S$ of given size $|T| = k$ has $\mathbf{E}_T[f(T)] \geq \frac{k}{|S|}f(S)$). Thus

$$\mathop{\mathbf{E}}_{\mathbf{y} \sim \mathrm{Ber}(\mathbf{x})}[\mathrm{SW}(\mathbf{y}, \mathbf{w}) - \mathrm{SW}(\mathbf{z}, \mathbf{w})] \leq \mathop{\mathbf{E}}_{\mathbf{y} \sim \mathrm{Ber}(\mathbf{x})}\left[\mathbb{I}\Big[|\mathbf{y}|_1 > k\Big] \cdot \frac{|\mathbf{y}|_1 - k}{|\mathbf{y}|_1} \cdot \mathrm{SW}(\mathbf{y}, \mathbf{w})\right].$$

Furthermore, as $\mathrm{OPT} = \max_{S \subseteq [n]:|S|=k} \mathrm{SW}(S, \mathbf{w})$ we have $\mathrm{OPT} \geq \mathbf{E}_{\mathbf{z} \sim \binom{y}{k}}[\mathrm{SW}(\mathbf{z}, \mathbf{w})] \geq \frac{k}{|\mathbf{y}|_1}\mathrm{SW}(\mathbf{y}, \mathbf{w})$ for each $\mathbf{y}$ with $|\mathbf{y}|_1 > k$. Therefore,

$$\mathop{\mathbf{E}}_{\substack{\mathbf{y} \sim \mathrm{Ber}(\mathbf{x}) \\ \mathbf{z} \sim \binom{y}{k}}}[\mathrm{SW}(\mathbf{y}, \mathbf{w}) - \mathrm{SW}(\mathbf{z}, \mathbf{w})] \leq \mathop{\mathbf{E}}_{\mathbf{y} \sim \mathrm{Ber}(\mathbf{x})}\left[\mathbb{I}\Big[|\mathbf{y}|_1 > k\Big] \cdot \frac{|\mathbf{y}|_1 - k}{k} \cdot \mathrm{OPT}\right]$$

$$= \frac{\mathrm{OPT}}{k} \cdot \sum_{i=1}^{n-k} i \cdot \mathop{\mathbf{Pr}}_{\mathbf{y} \sim \mathrm{Ber}(\mathbf{x})}[|\mathbf{y}|_1 = k + i] = \frac{\mathrm{OPT}}{k} \cdot \sum_{i=1}^{n-k} \mathop{\mathbf{Pr}}_{\mathbf{y} \sim \mathrm{Ber}(\mathbf{x})}[|\mathbf{y}|_1 \geq k + i].$$

CLAIM B.3. *Let $\mathbf{x} \in [0,1]^n$ with $|\mathbf{x}|_1 = k$. Then $\sum_{i \geq 1} \mathbf{Pr}_{\mathbf{y} \sim \mathrm{Ber}(\mathbf{x})}[|\mathbf{y}| \geq k + i] = O\big(\sqrt{k}\big)$.*

PROOF OF CLAIM B.3. By Chernoff bound, we have

$$\sum_{i=1}^{n-k} \mathop{\mathbf{Pr}}_{\mathbf{y} \sim \mathrm{Ber}(\mathbf{x})}[|\mathbf{y}| \geq k + i] \leq \sum_{i=1}^{n-k} e^{-\frac{i^2}{i+2k}} \leq \sum_{i=1}^{\infty} e^{-\frac{i^2}{i+2k}} \leq \int_0^{\infty} e^{-\frac{x^2}{x+2k}} \, \mathrm{d}x.$$

We further estimate this integral as

$$\int_0^{\infty} e^{-\frac{x^2}{x+2k}} \, \mathrm{d}x \leq \int_0^{\sqrt{k}} e^{-\frac{x^2}{x+2k}} \, \mathrm{d}x + \int_{\sqrt{k}}^{\infty} e^{-\frac{x}{1+2\sqrt{k}}} \, \mathrm{d}x \leq 1 \cdot \sqrt{k} - (1 + 2\sqrt{k}) \, e^{-\frac{x}{1+2\sqrt{k}}}\Big|_{\sqrt{k}}^{+\infty} = O(\sqrt{k}).$$

This concludes the proof of Claim B.3. $\square$

Claim B.3 allows us to conclude that $\mathbf{E}_{\mathbf{y} \sim \mathrm{Ber}(\mathbf{x})}[\mathrm{SW}(\mathbf{y}, \mathbf{w}) - \mathrm{SW}(\mathbf{z}, \mathbf{w})] = O\big(\frac{1}{\sqrt{k}}\big) \cdot \mathrm{OPT}$, i.e.,

$$\mathop{\mathbf{E}}_{\mathbf{y} \sim \mathrm{Ber}(\mathbf{x})}[\mathrm{SW}(\mathbf{z}, \mathbf{w})] = \mathrm{SW}(\mathbf{x}, \mathbf{w}) - O\left(\frac{1}{\sqrt{k}}\right) \cdot \mathrm{OPT} = \left(1 - 43 \, k^{-1/4} - O(k^{-1/2})\right)\mathrm{OPT}. \qquad \square$$

# C BETTER ALGORITHM FOR SINGLE-ITEM AUCTION

In Section 4, we studied the Bidder Selection Problem (BSP) for position auctions and obtained a $\left(1 - O(k^{-1/4})\right)$-approximate algorithm. In this section, we study the special case of single-item auction (i.e., $\ell$-unit auction with $\ell = 1$) as it was extensively studied in previous work, and give a better approximation ratio $\left(1 - O(\sqrt{\ln k/k})\right)$.

Similar to Section 4.1, we fix a small set $S_{\mathrm{fix}}$ of $\varepsilon \cdot k$ bidders, which affects the final approximation by at most $(1 - \varepsilon)$ factor due to the submodularity of BSP as a set function of the invited bidders. Formally, recall that for the single-item auction

$$\mathrm{SW}(\mathbf{x}) = \int_0^{+\infty} \mathrm{H}_{\mathrm{ber}}(\mathbf{q}(\mathbf{x}, \tau), 1) \, \mathrm{d}\tau = \int_0^{+\infty} \mathop{\mathbf{Pr}}_{\mathbf{v} \sim \mathbf{x} \cdot \mathbf{D}}[\exists v_i \geq \tau] \, \mathrm{d}\tau.$$

We find the set of $\varepsilon \cdot k$ bidders $S_{\mathrm{fix}}$ such that $\mathbf{Pr}_{\mathbf{v} \sim \mathbf{D}}[\exists i \in S_{\mathrm{fix}} : v_i \geq \eta] \geq 1 - \varepsilon$ for the largest possible threshold $\eta$. This allows us to take care of thresholds $\tau$ in a low range $\tau \in [0, \eta]$ by including $S_{\mathrm{fix}}$ in the solution (i.e., make $x_i = 1$ for all $i \in S_{\mathrm{fix}}$). Naturally, we want to pick bidders with higher probabilities $\mathbf{Pr}[v_i > \eta]$ into $S_{\mathrm{fix}}$, which means that for the high range thresholds $\tau > \eta$ we get the small probability property for each bidder $i \notin S_{\mathrm{fix}}$. This allows us to reduce BSP to the case of small probabilities tail events (Appendix A) for $\tau > \eta$ and bidders $i \in [n] \setminus S_{\mathrm{fix}}$, which can be effectively solved by the Poisson approximation. Formally, we can get the following guarantees for $S_{\mathrm{fix}}$.

CLAIM C.1 (SMALL BIDDER SET). *Let $\varepsilon \geq \sqrt{\frac{\ln k}{k}}$ be a multiple of $1/k$. We can find in polynomial time a threshold $\eta \geq 0$ and a set $S_{\mathrm{fix}} \subset [n]$ of size $|S_{\mathrm{fix}}| = \varepsilon \cdot k$, such that*

(a) $\displaystyle \mathop{\mathbf{Pr}}_{\mathbf{v} \sim \mathbf{D}}[\exists i \in S_{\mathrm{fix}} : v_i \geq \eta] \geq 1 - \frac{1}{k}$; (b) $\forall i \notin S_{\mathrm{fix}}, \displaystyle \mathop{\mathbf{Pr}}_{v_i \sim D_i}[v_i > \eta] < \varepsilon$.

PROOF. Recall that all distributions $\{D_i\}_{i \in [n]}$ have finite support. Thus we can search through all thresholds $\tau$ in polynomial time. There must be two consecutive threshold values $\eta$ and $\eta_+ > \eta$ such that the number of bidders $|\{i : \mathbf{Pr}[v_i \geq \eta] \geq \varepsilon\}| \geq \varepsilon \cdot k$ with large tail probabilities $\mathbf{Pr}[v_i \geq \eta] \geq \varepsilon$ is at least $\varepsilon \cdot k$, but a similar number of bidders $|\{i : \mathbf{Pr}[v_i > \eta] = \mathbf{Pr}[v_i \geq \eta_+] \geq \varepsilon\}| < \varepsilon \cdot k$ for the next threshold value $\eta_+$ is strictly less than $\varepsilon \cdot k$.

Let us place each bidder $i$ with $\mathbf{Pr}[v_i > \eta] \geq \varepsilon$ into $S_{\text{fix}}$ and fill the remaining positions in $S_{\text{fix}}$ up to size $\varepsilon \cdot k$ (so that $|S_{\text{fix}}| = \varepsilon \cdot k$) with bidders from $\{i : \mathbf{Pr}[v_i \geq \eta] \geq \varepsilon > \mathbf{Pr}[v_i \geq \eta_+]\}$. Then, every bidder $i \notin S_{\text{fix}}$ has $\mathbf{Pr}[v_i > \eta] = \mathbf{Pr}[v_i \geq \eta_+] < \varepsilon$ as required by condition (b). On the other hand, $|S_{\text{fix}}| = \varepsilon \cdot k$ and $\mathbf{Pr}[v_i \geq \eta] \geq \varepsilon$ for every $i \in S_{\text{fix}}$, i.e., condition (a) is satisfied since

$$\mathbf{Pr}_{\mathbf{v} \sim \mathbf{D}}[\exists i \in S_{\text{fix}} : v_i \geq \eta] \geq 1 - (1 - \varepsilon)^{\varepsilon \cdot k} \geq 1 - e^{-\varepsilon^2 \cdot k} \geq 1 - \frac{1}{k},$$

where to get the second inequality, we used the fact that $(1 - \frac{1}{x})^x < e^{-1}$ for any $x \geq 1$. □

After selecting such set $S_{\text{fix}}$ and threshold $\eta$, we are ready to give the complete description of Algorithm 4.

---

**Algorithm 4: Fractional BSP for Single-Item Auction.**

Fix $\varepsilon = \sqrt{\frac{\ln k}{k}}$ rounded up to a multiple of $1/k$, then do the following steps:

(1) Find $(\eta, S_{\text{fix}})$ as in Claim C.1. Set $x_i = 1$ for $\forall i \in S_{\text{fix}}$.

(2) For the remaining bidders $M \overset{\text{def}}{=} [n] \setminus S_{\text{fix}}$ let the Poisson approximation $\widetilde{\text{SW}}(\mathbf{x}_M)$ be

$$\widetilde{\text{SW}}(\mathbf{x}_M) \overset{\text{def}}{=} \left(1 - \frac{1}{k}\right)\eta + \int_\eta^{+\infty} \widetilde{\text{H}}_{\text{pois}}(\mathbf{q}(\mathbf{x}_M, \tau)) \, d\tau, \qquad \text{where} \qquad (12)$$

$$\widetilde{\text{H}}_{\text{pois}}(\mathbf{q}_M) \overset{\text{def}}{=} r_\tau + (1 - r_\tau) \cdot \text{H}_{\text{pois}}(\mathbf{q}_M, \ell = 1), \quad \text{and} \quad r_\tau \overset{\text{def}}{=} \mathbf{Pr}_{\mathbf{v} \sim \mathbf{D}}[\exists i \in S_{\text{fix}} : v_i \geq \tau]. \qquad (13)$$

(3) Return $\widetilde{\mathbf{x}}^* = (\mathbf{1}_{S_{\text{fix}}}, \widetilde{\mathbf{x}}_M^*)$, where $\widetilde{\mathbf{x}}_M^*$ is the solution to the concave program in $\mathbf{x}_M$:

$$\begin{aligned} \text{Maximize} \quad & \widetilde{\text{SW}}(\mathbf{x}_M) \\ \text{Subject To} \quad & \textstyle\sum_{i \in M} x_i \leq k - \varepsilon \cdot k, \quad x_i \in [0, 1] \quad \forall i \in M. \end{aligned} \qquad (14)$$

---

In the algorithm, we ignore thresholds $\tau \in [0, \eta]$, as by taking $S_{\text{fix}}$ we have already achieved high success probability of at least $1 - \frac{1}{k}$ by Claim C.1. For the high range thresholds $\tau > \eta$, we first observe that as the result of fixing set $S_{\text{fix}}$, the probability that there is a bidder with value greater than the threshold $\tau$ becomes

$$\text{H}_{\text{ber}}(\mathbf{q}, 1) = \mathbf{Pr}[\exists i \in [n] : v_i \geq \tau] = r_\tau + (1 - r_\tau) \cdot \mathbf{Pr}[\exists i \in M : v_i \geq \tau],$$

where $r_\tau$ is a constant that we can easily compute. Hence, we respectively adjust the Poisson approximation term $\widetilde{\text{H}}_{\text{pois}}(\mathbf{q}_M)$ in the algorithm according to (13) (we slightly abuse notations by writing $\text{H}_{\text{pois}}(\mathbf{q}_M)$ instead of $\text{H}_{\text{pois}}(\mathbf{q})$: for the coordinates $i \notin M$ we let $q_i = 0$).

THEOREM C.2. *Algorithm 4 for single-item auction works in polynomial time and is a $(1 - 2\varepsilon)$-approximation, i.e., a $\left(1 - O\left(\sqrt{\ln k/k}\right)\right)$-approximation to the fractional BSP.*

PROOF. We first verify that Algorithm 4 is polynomial. Note that the step (1) works in polynomial time by Claim C.1. For each $\tau$ in the support of $D_i$ we calculate in polynomial time the constants $r_\tau \in [0, 1]$. Both $\widetilde{\text{H}}_{\text{pois}}(\mathbf{q}_M)$ for each $\mathbf{x}_M$ and $\tau$ in the support and $\widetilde{\text{SW}}(\mathbf{x}_M)$ for each $\mathbf{x}_M$ can be computed in polynomial time. Moreover, all first and second order partial derivatives of $\widetilde{\text{SW}}(\mathbf{x}_M)$ can be computed in the same way as the integral of respective derivatives of $\widetilde{\text{H}}_{\text{pois}}(\mathbf{x}_M)$. Furthermore, it is easy to see that $\widetilde{\text{SW}}(\mathbf{x}_M)$ is a concave function in $\mathbf{x}_M$, since it is a positive linear combination of constant terms (such as $(1 - 1/k)\eta$ and $r_\tau$) and concave functions $\text{H}_{\text{pois}}(\mathbf{x}_M)$ according to Claim 4.2. Hence, we can find the optimal solution $\widetilde{\mathbf{x}}_M^*$ in polynomial time using standard concave (first or second order) maximization methods.

To prove an approximation guarantee of $1 - 2\varepsilon$ for Algorithm 4, we first derive the following approximations of $\text{SW}(\mathbf{x}_M, \mathbf{1}_{S_{\text{fix}}})$ by $\widetilde{\text{SW}}(\mathbf{x}_M)$ similar to (11) (but in a special case of $\ell = 1$).

LEMMA C.3. $\forall \mathbf{x}_M \in [0, 1]^{|M|}, 0 \leq \text{SW}(\mathbf{x}_M, \mathbf{1}_{S_{\text{fix}}}) - \widetilde{\text{SW}}(\mathbf{x}_M) \leq \varepsilon \cdot \text{SW}(\mathbf{x}_M, \mathbf{1}_{S_{\text{fix}}})$.

PROOF. Recall that by (3) the Social Welfare $\text{SW}(\mathbf{x})$ for $\mathbf{x} = (\mathbf{x}_M, \mathbf{1}_{S_{\text{fix}}})$ and $\ell = 1$ is

$$\text{SW}(\mathbf{x}) = \int_0^\eta \text{H}_{\text{ber}}(\mathbf{q}(\mathbf{x}, \tau)) \, d\tau + \int_\eta^{+\infty} \text{H}_{\text{ber}}(\mathbf{q}(\mathbf{x}, \tau)) \, d\tau.$$

For the low range $\tau \in [0, \eta]$ we have $\text{H}_{\text{ber}}(\mathbf{q}(\mathbf{x}, \tau)) \in [1 - \frac{1}{k}, 1]$ due to the choice of $S_{\text{fix}}$ in Claim C.1. It is well approximated by the respective term $\left(1 - \frac{1}{k}\right)\eta$ in (12).

For the high range $\tau > \eta$, we apply Lemma A.2 (b) with $\delta = \varepsilon$ and get the following bound:

$$H_{ber}(\mathbf{q}) - \widetilde{H}_{pois}(\mathbf{q}) = (1 - r_\tau)\Big(H_{ber}(\mathbf{q}_M) - H_{pois}(\mathbf{q}_M)\Big) \leq (1 - r_\tau)\varepsilon \cdot H_{ber}(\mathbf{q}_M) \leq \varepsilon \cdot H_{ber}(\mathbf{q}).$$

On the other hand, $H_{ber}(\mathbf{q}) - \widetilde{H}_{pois}(\mathbf{q}) \geq 0$, as $H_{ber}(\mathbf{q}_M) \geq H_{pois}(\mathbf{q}_M)$ by Lemma A.2 (b). Thus

$$0 \leq SW(\mathbf{x}) - \widetilde{SW}(\mathbf{x}_M) \leq \frac{\eta}{k} + \varepsilon \cdot \int_\eta^{+\infty} H_{ber}(\mathbf{q}(\mathbf{x}, \tau)) \, d\tau \leq \max\left(\frac{1}{k-1}, \varepsilon\right) \cdot SW(\mathbf{x}) = \varepsilon \cdot SW(\mathbf{x}),$$

where the last equality holds since $\varepsilon \geq \sqrt{\frac{\ln k}{k}}$. □

Now we are ready to complete the proof of Theorem C.2. Let $\mathbf{x}^*$ be the optimal solution to fractional BSP. We consider $\mathbf{x}_+^* \in \mathbb{R}_{\geq 0}^n$ defined as $\mathbf{x}_+^* \overset{\text{def}}{=} (\mathbf{x}_M^*, \mathbf{1}_{S_{fix}})$, so that $\mathbf{x}_+^* \geq \mathbf{x}^*$. Then, by Lemma C.3 for $\mathbf{x} = \mathbf{x}_+^*$ we have

$$\widetilde{SW}(\mathbf{x}_M^*) \geq (1 - \varepsilon) \cdot SW(\mathbf{x}_+^*) \geq (1 - \varepsilon) \cdot SW(\mathbf{x}^*).$$

On the other hand, by Lemma C.3 for $\mathbf{x} = \widetilde{\mathbf{x}}^*$ we have

$$SW(\widetilde{\mathbf{x}}^*) \geq \widetilde{SW}(\widetilde{\mathbf{x}}_M^*) \geq \widetilde{SW}\left(\frac{k - \varepsilon \cdot k}{k} \cdot \mathbf{x}_M^*\right) \geq (1 - \varepsilon) \cdot \widetilde{SW}(\mathbf{x}_M^*) \geq (1 - 2\varepsilon) \cdot SW(\mathbf{x}^*),$$

where the second inequality holds, as $\widetilde{\mathbf{x}}_M^*$ is the optimal solution to (14) and $\frac{k - \varepsilon \cdot k}{k} \cdot \mathbf{x}_M^*$ is a feasible solution; the third inequality holds, as $\widetilde{SW}(\mathbf{x})$ is a concave function in $\mathbf{x}$ by Claim 4.2; the last inequality holds, as $\widetilde{SW}(\mathbf{x}_M^*) \geq (1 - \varepsilon) \cdot SW(\mathbf{x}^*)$ and $(1 - \varepsilon)^2 \geq 1 - 2\varepsilon$. This concludes the proof. □

# D COMPARING OUR THEORETICAL AND MODIFIED ALGORITHMS

In this section, we compare the performance of the theoretical version and the modified version of our algorithm. Due to certain limitations on the convex objectives in Gurobi, we implemented the theoretical version of our algorithm in MATLAB with the help of another convex optimization library, Mosek [1], and ran it on the same set of test inputs as in Section 5. As the implementations are in different programming languages, we do not compare their running time and only compare their approximation efficiency.

**Table 2: Experimental results of Local Search, Greedy, and both the modified version and the theoretical version of our algorithm. For each algorithm, we show the average relative quality of the produced solution to that of the best-performing algorithm that terminated in 1 week. Error bars denote the standard deviation.**

| Setting | | Benchmarks | | Our Algorithm | |
|---|---|---|---|---|---|
| $n$ | $k$ | Local Search | Greedy | Modified | Theoretical |
| | 5 | 100.00% ± 0.00% | 98.93% ± 0.50% | 99.99% ± 0.03% | 95.28% ± 1.10% |
| 50 | 10 | 100.00% ± 0.00% | 98.71% ± 0.38% | 99.99% ± 0.04% | 97.43% ± 0.84% |
| | 20 | 100.00% ± 0.00% | 99.17% ± 0.28% | 99.99% ± 0.01% | 99.53% ± 0.23% |
| | 10 | 100.00% ± 0.00% | 98.06% ± 0.35% | 99.99% ± 0.01% | 95.36% ± 0.79% |
| 200 | 20 | 100.00% ± 0.00% | 97.92% ± 0.24% | 100.00% ± 0.00% | 98.50% ± 0.45% |
| | 40 | 100.00% ± 0.00% | 97.97% ± 0.26% | 99.99% ± 0.00% | 99.85% ± 0.10% |
| | 50 | 100.00% ± 0.00% | 97.18% ± 0.14% | 99.99% ± 0.00% | 99.80% ± 0.09% |
| 1000 | 100 | N/A | 97.15% ± 0.15% | 100.00% ± 0.00% | 99.96% ± 0.02% |
| | 200 | N/A | 97.38% ± 0.11% | 100.00% ± 0.00% | 99.96% ± 0.02% |

As shown in Table 2, we can see that the theoretical version of our algorithm performs slightly worse than the modified version. This is due to the potentially suboptimal decision of fixing a small bidder set $S_{fix}$. This step is helpful when we analyze our algorithm theoretically, but it may not be optimal in practice. Therefore, we choose to use the modified version of our algorithm in Section 5.

