# OpenReview forum: "Bidder Selection Problem in Position Auctions: A Fast and Simple Algorithm via Poisson Approximation"
_ACM.org/TheWebConf/2024/Conference — TheWebConf24 Oral_

### Official Review · Reviewer_2cGh · 2023-11-23

**Novelty:** 6
**Technical Quality:** 7

**Review:**

The paper studies the bidder selection problem where we choose a subset of k bidders out of n bidders to maximize an objective under a pre-determined auction format. In particular, the paper considers the welfare and revenue objectives and VCG and Myerson’s auction in a position auction environment.

Strengths:
1) The problem is motivated by practical challenges faced by ad platforms where the platforms may need to reduce the set of all available advertisers to a few advertisers before running complicated ML models and then an auction. The bidder selection problem is modeling the reduction/selection part.

2) The paper proposes a Poisson relaxation approach and provides an Efficient PTAS for a position auction environment. The position auction environment was not considered before by previous works.

3) The paper highlights the implementability of the proposed algorithm compared to other PTAS in previous works. The paper provides an empirical study for the proposed algorithm. The proposed algorithm is shown to perform well against two heuristic algorithms in both the running time and solution quality.

**Questions:**

None

**Reviewer Confidence:**

3: The reviewer is confident but not certain that the evaluation is correct

**Scope:**

4: The work is relevant to the Web and to the track, and is of broad interest to the community

---

### Official Review · Reviewer_6Z74 · 2023-11-24

**Novelty:** 4
**Technical Quality:** 5

**Review:**

This paper studies the bidder selection problem (BSP), where a large pool of n advertisers competes for ad slots.  The goal is to efficiently find a subset of k out of n bidders to optimize the expected social welfare or revenue, given n independent prior distributions. The authors study BSP in the multi-position VCG auctions for welfare and revenue optimization. They propose a Poisson relaxation of BSP and show that BSP is polynomial-time solvable up to a vanishing error as k grows. Also, the combination of the relaxation with the brute force algorithm yields a PTAS. Experiments show that the proposed algorithms outperforms Greedy in both running time and approximation on medium and large size instances.

Pros
* This paper proposes a fractional relaxation of BSP, and bounds the loss of rounding to integral results.
* Experiments show comparable quality and greatly improved running time compared to the benchmarks.

Cons
* According to Corollary 1.1, the worst case running time of the proposed algorithm is in $n^{poly(1/\varepsilon)}$, which looks much worse than previous work. It seems this paper shows the efficiency of their algorithm mainly by experiments. It is unfortunate that the problem is NP-hard, so they could only compare the results with some benchmark algorithms, but not the true optimal solution, which makes the conclusion weaker.

**Questions:**

In Table 1, the “solution” column denotes the relative quality of the solution to the best-performing algorithm. Is Local Search the “best performing algorithm”? Is it possible to actually calculate the optimal solution for smaller n and k and compare the results?

**Reviewer Confidence:**

2: The reviewer is willing to defend the evaluation, but it is likely that the reviewer did not understand parts of the paper

**Scope:**

3: The work is somewhat relevant to the Web and to the track, and is of narrow interest to a sub-community

---

### Official Review · Reviewer_MijP · 2023-11-25

**Novelty:** 6
**Technical Quality:** 7

**Review:**

I had previous reviewed this for another conference. I will paste my summary from the past below the line:

Assesment:
The paper addresses a very important problem that is faced by ad-auctions around the world. The authors propose a relatively simple mechanism which performs has strong theoretical guarantees and performs well on experiments. Although more efficient than prior work on the literature, I imagine it would still need to simpler to be put to use in practice. Nonetheless I view the simplicity of the idea as a feature rather than a bug, especially given the importance of the problem. The main ideas in the paper are cleanly presented and the algorithm and implementation are relatively straightforward.

One important caveat is that the paper doesn't address the incentive question but strictly tackles the optimization question. However, I think their contributions are novel and would be very interesting to the community. Therefore I feel this paper meets the bar for WWW 24



----
**Problem**
The authors study the bidder selection problem, a problem that has been studied under multiple names
in different contexts. Essentially, one must select $k$ from $n$ different bidders whose bids are
sampled independently from known distribution $(D_i)_{i \in [n]}.$ The platform will subsequently
run  an position auction (the common form for ad auctions in the internet) among the $k$ to maximize
its own  revenue.

The authors results follow from a new relaxation  with a concave objective that can be solved in polynomial(in $n$ and $k$) time. This relaxation is efficiently solvable and converges quickly as they show on a number of synthetic datasets. Finally, they show that they can get a $O(1-O(k^{1/4})$-approximation in general and a $1-O(\sqrt(\ln k/k)$-approximation for the single item case.


**Techniques**
The authors begin with the fractional relaxation of the problem where one is allowed to choose a fractional bidder with probability $x_i$. I will highlight the techniques for the case where the platform has $\ell$ items and the idea for the general position auction is very similar. Technically, one must maximize
$SW(x) := \mathbb{E}_{v\sim x\cdot D_i} \sum_{i=1}^{\ell} v_{(i)}
$
 where $v_{(i)}$ is the $i$-th largest value.
The authors first rewrite the problem as
\begin{align}
SW(x) = \int_{0}^{\infty} \mathbb{E}_{v\sim x\cdot D} \left[ \min \left(\sum_{i=1}^n \mathbb{I}[v_i \geq \tau  ], \ell \right) \right]  d \tau
\end{align}
and relax it by switching the expectation and minimization.
\begin{align}
\tilde{SW}(x) = \int_{0}^{\infty} \min \left( \mathbb{E}_{v\sim x\cdot D} \left[ \sum_{i=1}^n \mathbb{I}[v_i \geq \tau ]  \right]  , \ell \right)  d\tau
\end{align}
Note that $\tilde{SW}$ is a concave function which can be approximated when $x$ is allowed to be fractional. In fact they show that when $\ell$ is large the above approximation is close to $SW(x)$. However, when $\ell$ is small and the bulk of value of $SW(x)$ arrives from tail events, the authors approximate the min as a Poisson approximation. This also makes the objective to be concave and one that can be efficiently approximated. They show that when tail events dominate, the Poisson approximation is a good approximation the original objective.

Of course tail events may not always dominate. To get around this, they choose a small set of $\epsilon k$ bidders and a threshold such that with high probability that they will exceed the threshold. Now the remaining bidders will have to exceed the threshold with a small probability when the Poisson approximation applies.  This is a technique inspired by previous work of [17].


**Experiments**

The authors also implement their algorithm on three synthetic datasets (3-point distributions, normal, shifted sutdent's t-distribution). They show that their algorithm achieves near optimal results in all cases, often far exceeding the theoretical guarantees. This is not surprising and dovetails with previous observation of [17]. It is not clear to me that synthetic datasets are the hardest for this problem as the real life datasets may not obey prior assumptions as the authors claim. Nonetheless this is a useful tid-bit.

**Questions:**

Have the authors explored whether this relaxation or any modification allows for an incentive compatible mechanism for bidder selection?

**Reviewer Confidence:**

4: The reviewer is certain that the evaluation is correct and very familiar with the relevant literature

**Scope:**

4: The work is relevant to the Web and to the track, and is of broad interest to the community

---

### Official Review · Reviewer_HfGE · 2023-11-28

**Novelty:** 6
**Technical Quality:** 7

**Review:**

Summary

This paper studies the following problem: given a collection of n distributions over positive real numbers, how can you select a subset of k of them to maximize their expected maximum (i.e., the maximum of k independently drawn samples, one per chosen distribution). This is a very important problem in the design of modern ad auctions, which often proceed in two stages -- a preliminary "eligibility" stage where a large set of bidders is filtered down to a small set of eligible bidders, and a secondary auction stage where ML models estimate the value of a click for each of the remaining bidders (the eligibility stage is necessary since the models are too costly to run on all bidders). Different variants of this problem (i.e. maximizing the sum of the ell largest samples, or a weighted linear combination of the samples in sorted order) correspond to optimization problems faced by different auctions.

This paper proposes a new theoretical and empirical approach for solving this problem, based on the following two ideas. First, they consider a fractional relaxation of the problem,where picking fraction p of a distribution corresponds to scaling it down by p. Secondly (and more interestingly) they perform a further relaxation they call "Poisson relaxation", where they replace a sum of Bernoulli variables that appears in the objective of this problem with a sum of Poisson rvs with the same mean. These two relaxations reduce the problem to a convex problem, which can be solved in polynomial time with standard convex programming techniques (and some additional cleverness to ensure that these relaxations do not lose too much in approximation). Altogether, the authors obtain a (1 - O(k^{1/4}))-approximation to OPT in polynomial time.

They complement these results with an empirical investigation where they simulate a slightly simplified version of this algorithm on synthetically generated position auction data, and compare it to local search and submodular-greedy benchmarks. Their algorithm obtains the same quality solution as both benchmarks while running in substantially less time.

Evaluation

The problem the authors call the "Bidder Selection Problem" is a fundamental problem in modern auction design. The authors make some non-trivial theoretical observations about this problem, and present a new algorithm for it that has nice theoretical guarantees and is (mostly -- see below) feasible to implement practically. The paper was well-written and easy to read, and I think would be of interest to a good portion of the WebConf audience (especially the more theoretical crowd). Overall I am in favor of acceptance.

One gripe about the empirical section: I think it is worth pointing out that although the implementation they have of their algorithm is substantially faster than the submodular-greedy and local search benchmarks they compare to, the times they obtain (up to ~40 seconds for a pool of n = 1000 bidders) are probably still prohibitively slow for being used in the presented applications (e.g. search position auctions). I understand the authors have probably not super-optimized their approach in terms of speed, but there is a several order of magnitude gap between 40 seconds and what would be tolerable in practice. From an empirical point of view, it would be interesting to compare the quality of their algorithm to some even more naive benchmarks (e.g. taking the the distributions with the largest means / medians / quantiles); ideally there should be large observable gains in performance to justify this.

**Questions:**

Feel free to respond to any comments / potential misunderstandings in the review above.

**Reviewer Confidence:**

3: The reviewer is confident but not certain that the evaluation is correct

**Scope:**

3: The work is somewhat relevant to the Web and to the track, and is of narrow interest to a sub-community

---

### Decision · Program_Chairs · 2024-01-22

**Decision:**

Accept (Oral)

**Comment:**

The paper considers a multi-stage approach to ad auctions where (due to computational resources) in the first stage the advertising platform selects k ads out of n, and in the second stage an auction is run for those k ads. The authors propose a Poisson relaxation of the problem which yields a polynomial time algorithm, and they combine it with a brute-force search for a PTAS for position auctions.

 The reviewers appreciated the relevance of the problem, the technical contributions of the paper, and the clarity of the writing, and as such recommend it for acceptance.